# Ubiquitinated-PCNA protects replication forks from DNA2-mediated degradation by regulating Okazaki fragment maturation and chromatin assembly

Tanay Thakar [1], Wendy Leung[2], Claudia M. Nicolae[1], Kristen E. Clements[1], Binghui Shen[3], Anja-Katrin Bielinsky [2] & George-Lucian Moldovan [1✉]

Upon genotoxic stress, PCNA ubiquitination allows for replication of damaged DNA by recruiting lesion-bypass DNA polymerases. However, PCNA is also ubiquitinated during normal S-phase progression. By employing 293T and RPE1 cells deficient in PCNA ubiquitination, generated through CRISPR/Cas9 gene editing, here, we show that this modification promotes cellular proliferation and suppression of genomic instability under normal growth conditions. Loss of PCNA-ubiquitination results in DNA2-dependent but MRE11-independent nucleolytic degradation of nascent DNA at stalled replication forks. This degradation is linked to defective gap-filling in the wake of the replication fork and incomplete Okazaki fragment maturation, which interferes with efficient PCNA unloading by ATAD5 and subsequent nucleosome deposition by CAF-1. Moreover, concomitant loss of PCNA-ubiquitination and the BRCA pathway results in increased nascent DNA degradation and PARP inhibitor sensitivity. In conclusion, we show that by ensuring efficient Okazaki fragment maturation, PCNA-ubiquitination protects fork integrity and promotes the resistance of BRCA-deficient cells to PARP-inhibitors.

[1] Department of Biochemistry and Molecular Biology, The Pennsylvania State University College of Medicine, Hershey, PA 17033, USA. [2] Department of Biochemistry, Molecular Biology and Biophysics, College of Biological Sciences, University of Minnesota, Minneapolis, MN 55455, USA. [3] Department of Cancer Genetics and Epigenetics, Beckman Research Institute of City of Hope, Duarte, CA 91010, USA. ✉email: glm29@psu.edu

Accurate DNA replication is essential for genomic stability[1,2]. DNA replication is initiated at discrete replication origins and occurs in a continuous manner on the leading strand, catalyzed by DNA polymerase Polε. In contrast, lagging strand replication needs frequent re-priming by the Polα–primase complex, followed by processive DNA synthesis by Polδ. This results in short RNA-primed DNA fragments known as Okazaki fragments (OFs). An essential component of the replication machinery is the homotrimeric ring-shaped protein proliferating cell nuclear antigen (PCNA), which encircles and slides along the DNA during DNA synthesis. PCNA is loaded at replication origins by the RFC1–5 complex, and unloaded upon replication termination by an alternative complex in which ATAD5 (Elg1 in yeast) replaces RFC1[3,4]. During DNA synthesis, PCNA interacts with the replicative polymerases on each strand and enhances their processivities[5,6]. On the lagging strand PCNA recruits the Flap endonuclease (FEN1) which cleaves the RNA primer displaced by Polδ, and DNA ligase 1 (LIG1) which seals the resulting nick to complete OF maturation (OFM)[7]. Concomitant with DNA replication, PCNA promotes chromatinization of the newly synthesized DNA by recruiting the chromatin assembly factor CAF-1 and other histone chaperones[8,9]. These interactions are mediated by a motif termed PCNA-interacting peptide (PIP)-box[5,10].

Unrepaired DNA lesions, secondary DNA structures, and other difficult to replicate sequences, can induce the arrest of the replicative polymerases, causing replication stress[11,12]. In response to replication stress, PCNA is mono-ubiquitinated by the RAD18 ubiquitin ligase at lysine 164 (K164). This modification promotes a switch from the replicative polymerase to specialized low-fidelity polymerases, which preferentially bind ubiquitinated-PCNA ($^{Ubi}$PCNA)[5,6,13–17]. These polymerases bypass replication obstacles to ensure efficient DNA replication, a process known as translesion synthesis (TLS)[18,19].

In response to replication stress, forks can be reversed into four-way junctions upon annealing of the complementary nascent strands. Fork reversal protects against fork collapse and provides an opportunity to bypass the DNA injury by using the nascent strand of the sister chromatid as a temporary template[20–22]. However, reversal can also render replication forks susceptible to nucleolytic processing. In cells lacking functional BRCA pathway, reversed forks are subject to resection by the nuclease MRE11[23,24]. This degradation drives genome instability and may underlie the sensitivity of BRCA-mutant cells to cisplatin and PARP inhibitors (PARPi) such as olaparib[25,26]. In addition to MRE11, other nucleases including DNA2, EXO1, CTIP, and MUS81 have been implicated in resection of nascent DNA at stalled forks and subsequent genome instability[27–30].

In vertebrate cells, mono-ubiquitination is the prevalent form of modified PCNA, although poly-ubiquitination can also be detected[31–33]. While PCNA ubiquitination is induced upon replication stress, basal levels of mono-ubiquitinated PCNA can be detected in S-phase cells under unperturbed growth conditions[16,31,32]. This suggests that, in human cells, PCNA ubiquitination may play an important but so far elusive role in controlling replication fork progression and genome stability during normal S-phase. Here, we show that loss of PCNA ubiquitination renders nascent DNA at stalled replication forks susceptible to degradation by the nuclease DNA2. Mechanistically, we link this nucleolytic degradation to inefficient gap-filling in the wake of the replication fork which interferes with efficient Okazaki fragment ligation, precluding PCNA unloading by ATAD5 and subsequent nucleosome deposition by CAF-1. We therefore define the $^{Ubi}$PCNA–LIG1–ATAD5–CAF-1 genetic pathway of replication fork protection that operates in parallel to the BRCA–RAD51 pathway.

## Results

**Generation of PCNA–K164R mutant cells.** As PCNA is essential for cell proliferation, previous studies investigating the role of PCNA ubiquitination in human cell lines relied on siRNA-mediated depletion of endogenous PCNA coupled with transfection of a K164R mutant or PCNA-ubiquitin fusion peptides[6]. However, the residual expression from the endogenous PCNA locus and the artificial overexpression of the PCNA variants can complicate the analyses. In order to overcome these limitations, we employed CRISPR/Cas9 gene editing to introduce the K164R homozygous mutation in the endogenous PCNA gene, in 293T and RPE1 cell lines. Monoclonal cultures were initially screened for loss of PCNA ubiquitination by western blot using an antibody specific for ubiquitinated PCNA. Several K164R mutant 293T clones were obtained. However, the level of unmodified PCNA in these clones was reduced compared to the parental line, as shown for the clone KR5, in Supplementary Fig. 1a. The genome of 293T cells is considered pseudotriploid[34]. Cloning of individual PCNA alleles from the KR5 cell line followed by Sanger sequencing revealed, in addition to PCNA-K164R allele(s), other alleles in which the PCNA gene was inactivated through introduction of small insertions or deletions. To exclude phenotypes caused by reduced PCNA expression and off-target effects of CRISPR/Cas9, we created an isogenic pair by re-expressing wildtype PCNA or the K164R mutant in the KR5 clone through a lentiviral expression system. The resulting cell lines, termed 293T-WT and 293T-K164R (or KR from here on), show similar levels of unmodified PCNA between themselves and when compared to the parental cell line (Fig. 1a; Supplementary Fig. 1a). In contrast to 293T cells, RPE1 cells are near-diploid[35]. Two RPE1 clones harboring the PCNA-K164R mutation endogenously were generated (Supplementary Fig. 1b). Both clones showed similar levels of unmodified PCNA as the parental line. Sequencing of the genomic region confirmed that, in both clones, both PCNA alleles were homozygously edited with the desired mutation, and thus they were used for subsequent experiments without the complementation employed for 293T cells.

**Replication stress and increased fork speed in K164R cells.** As expected from the well-established role of PCNA in TLS, both 293T-K164R and RPE1-K164R cells were sensitive to DNA damaging agents that induce single-stranded DNA lesions, such as UV and cisplatin (Fig. 1b–e). Moreover, 293T-K164R cells showed reduced UV-induced mutagenesis rates (Supplementary Fig. 1c) as measured by the SupF shuttle vector assay[36]—in line with the role of PCNA ubiquitination in recruiting the TLS polymerase Polη to bypass UV-induced lesions[16]. Rather unexpectedly, however, under unperturbed growth conditions, KR clones showed lower proliferation rates (Supplementary Fig. 1d), and a reduced proportion of cells undergoing DNA synthesis (Supplementary Fig. 1e). As this pattern was reminiscent of cells experiencing increased levels of endogenous replication stress[37], we next investigated expression of DNA damage markers. We observed that, under normal growth conditions, KR cells showed increased levels of CHK2 phosphorylation (Fig. 1f) and 53BP1 chromatin foci (Fig. 1g, h), indicating DNA damage accumulation. These findings suggest that PCNA ubiquitination-deficient cells are unable to resolve endogenous replication stress, resulting in DNA damage accumulation under unperturbed growth conditions.

To evaluate the role of PCNA ubiquitination in replication fork progression, we employed the DNA fiber combing assay to measure replication dynamics in K164R cells. Under unperturbed growth conditions, both 293T-K164R and RPE1-K164R cells showed longer nascent tract length and increased replication fork

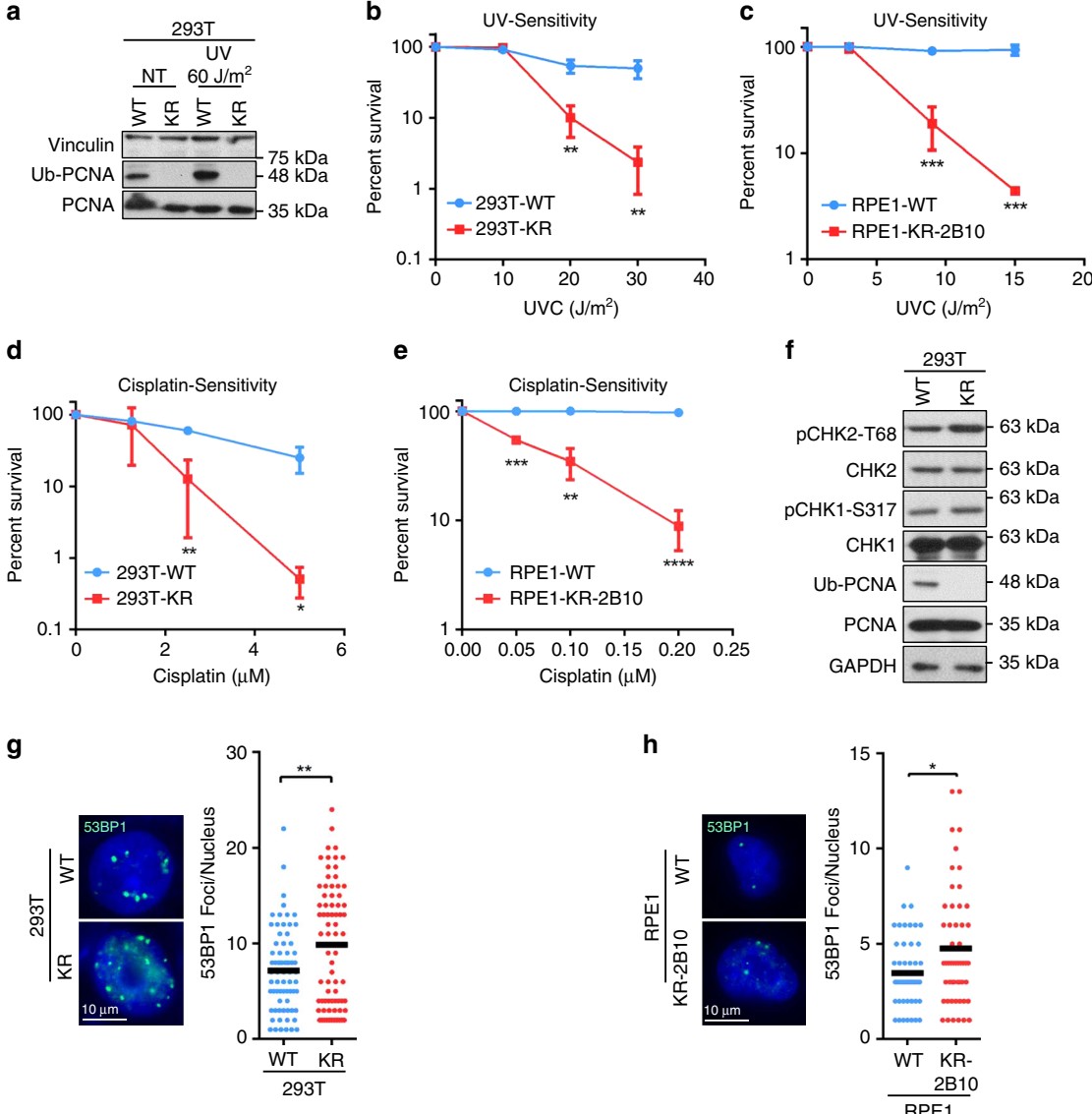

**Fig. 1 PCNA ubiquitination suppresses accumulation of endogenous DNA damage. a** Western blot experiment showing the loss of PCNA ubiquitination in 293T-K164R cells generated through CRISPR/Cas9 gene editing. Denatured whole cell extracts of cells under normal growth conditions, or 3 h after exposure to the indicated UV dose, were analyzed. A similar experiment performed in RPE1-K164R cells is shown in Supplementary Fig. 1b. **b–e** Clonogenic survival experiments showing hypersensitivity of 293T-K164R **b**, **d** and RPE1-K164R **c**, **e** cells to UV **b**, **c** and cisplatin **d**, **e**. The average of three experiments, with standard deviations indicated as error bars, is shown. Asterisks indicate statistical significance (t-test, two-tailed, unequal variance). **f** Western blot experiment showing increased CHK2 phosphorylation in 293T-K164R cells under normal growth conditions. **g**, **h** Immunofluorescence experiment showing increased 53BP1 chromatin foci in unsynchronized 293T-K164R **g** and RPE1-K164R **h** cells. At least 50 cells were quantified for each condition. The mean values are marked on the graph, and asterisks indicate statistical significance (t-test, two-tailed, unequal variance). Representative micrographs are also shown. Source data are provided as a Source Data file.

speed (Fig. 2a, b; Supplementary Fig. 2a, b). Previously, it was shown that the ZRANB3 translocase is recruited by K63-linked polyubiquitinated PCNA to mediate slowing of replication forks in the presence of replication stress[38]. In line with this, we found that 293T-K164R cells were unable to efficiently reduce fork speed in the presence of low levels (0.4 mM) of the replication fork stalling agent hydroxyurea (HU) (Supplementary Fig. 2c). This raises the possibility that the longer nascent tracts observed in KR cells under normal growth conditions may simply reflect the loss of ZRANB3 recruitment to stressed replication forks. To address this, we depleted ZRANB3 in wildtype cells. This did not result in longer nascent tracts (Supplementary Fig. 2d, e), arguing against a role for ZRANB3-mediated fork slowing in controlling fork speed under normal growth conditions.

**PCNA ubiquitination protects stalled forks from degradation.** To investigate if the abnormal replication fork characteristics described above are associated with defects in fork stability, we measured replication fork integrity in the presence of acute replication stress. Treatment with 4 mM HU resulted in degradation of the nascent DNA tract in both 293T-K164R and RPE1-K164R cells, but not in the respective control cells (Fig. 2a–d). Nascent strand degradation was observed under two different experimental conditions: when HU was added for 3 h in between the IdU and CldU pulses and the IdU tract length was measured (Fig. 2a, b), and when HU was added for 4.5 h after consecutive incubations with thymidine analogs and the ratio of CldU to IdU tract-lengths was calculated (Fig. 2c, d). HU-induced nascent strand degradation has been extensively described in the context

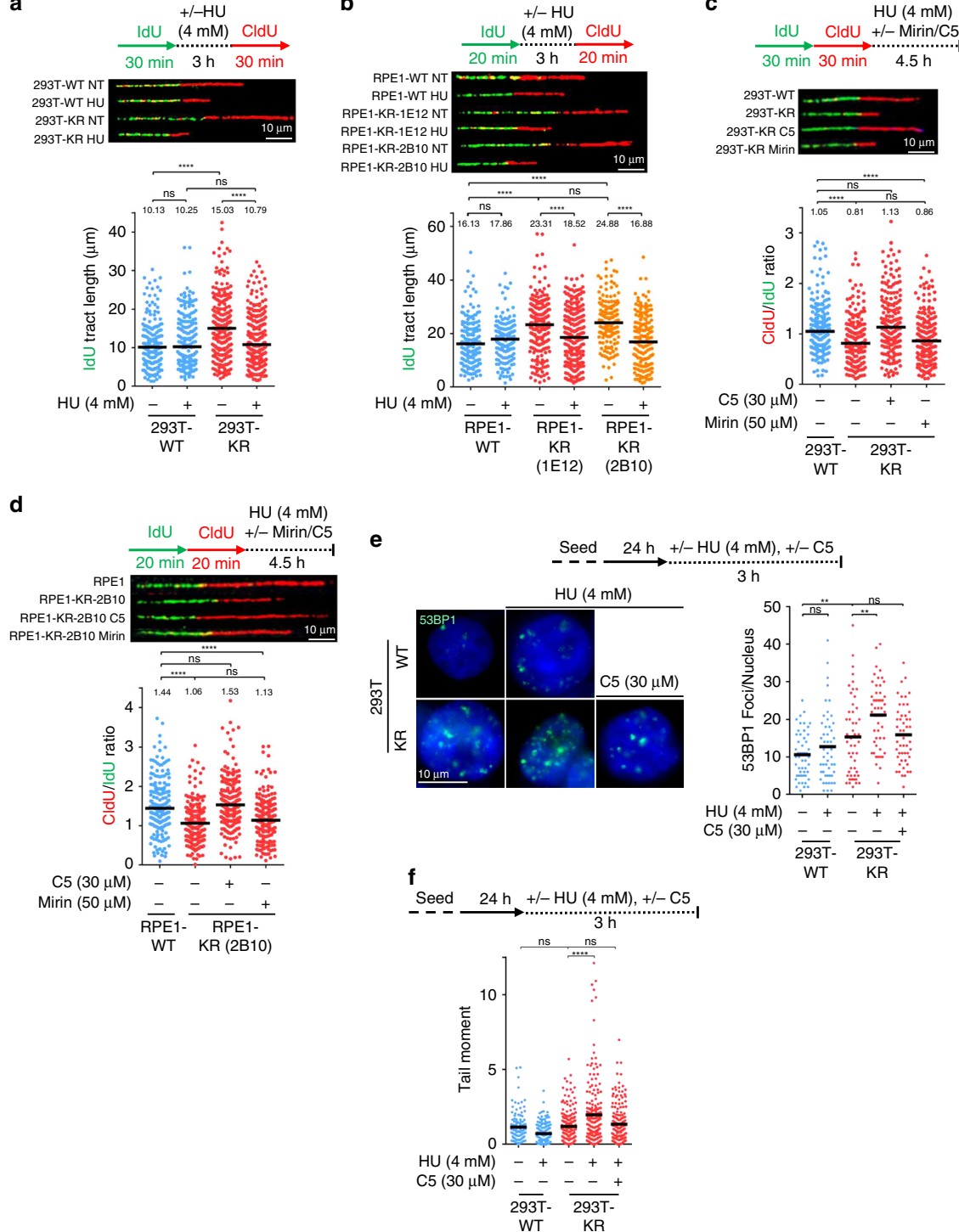

**Fig. 2 PCNA-K164R cells exhibit DNA2-mediated nascent DNA degradation. a, b** DNA fiber combing assays showing faster replication fork progression in 293T-K164R cells **a** and two different clones of RPE1-K164R cells **b** under normal growth conditions, and nascent strand degradation upon HU treatment. The quantification of the IdU tract length is presented, with the median values marked on the graph and listed at the top. Asterisks indicate statistical significance (Mann–Whitney test, two-sided). Schematic representation of the assay conditions, and representative micrographs are also presented. **c, d** HU-induced nascent strand degradation in 293T-K164R **c** and RPE1-K164R **d** cells is suppressed by incubation with the DNA2 inhibitor C5, but not by treatment with the MRE11 inhibitor mirin. The ratio of CldU to IdU tract lengths is presented, with the median values marked on the graph and listed at the top. Asterisks indicate statistical significance (Mann–Whitney test, two-sided). A schematic representation of the assay conditions, as well as representative micrographs are also presented. **e** Immunofluorescence experiment showing that HU treatment augments 53BP1 foci formation in unsynchronized 293T-K164R cells. DNA2 inhibition suppresses 53BP1 foci formation in KR cells. At least 50 cells were quantified for each condition. The mean values are marked on the graph, and asterisks indicate statistical significance (t-test, two-tailed, unequal variance). Representative micrographs are also shown. **f** Neutral comet assay showing that 293T-K164R cells accumulate DSBs upon HU treatment. DNA2 inhibition suppresses this accumulation. At least 100 cells were quantified for each condition. The mean values are marked on the graph, and asterisks indicate statistical significance (t-test, two-tailed, unequal variance). Source data are provided as a Source Data file.

of BRCA deficiency where it is dependent on the activity of the MRE11 nuclease[21,23,24]. Surprisingly, MRE11 inhibition using the inhibitor mirin did not suppress nascent strand degradation in KR cells (Fig. 2c, d; Supplementary Fig. 3a–c), indicating that a different fork degradation pathway operates in these cells. We further ruled out the involvement of other nucleases previously involved in nascent strand degradation, including EXO1, CTIP, and MUS81[29,30] (Supplementary Fig. 3d, e). In contrast, inhibition of the nuclease DNA2 with the specific inhibitor C5, or siRNA-mediated knockdown of DNA2, completely restored nascent tract length in both 293T-K164R and RPE1-K164R cells (Fig. 2c, d; Supplementary Fig. 3f, g), indicating that DNA2 is the nuclease responsible for fork degradation upon loss of PCNA ubiquitination. The WRN helicase has been previously described as a cofactor for DNA2 in nascent strand degradation[27]. In line with this, WRN depletion also rescued HU-induced fork degradation in KR cells (Supplementary Fig. 3f, g).

The K164 residue of PCNA is subjected not only to ubiquitination, but also to SUMOylation[39,40]. Depletion of the ubiquitin ligase RAD18 recapitulated the PCNA-K164R phenotype, as it resulted in DNA2-mediated nascent strand degradation (Supplementary Fig. 4a–c). Depletion of the SUMO-conjugating enzyme UBC9 did not affect fork stability (Supplementary Fig. 4a, b). Finally, depletion of the ubiquitin ligase UBC13, involved in PCNA poly-ubiquitination by K63-linked ubiquitin chains[5,15], also resulted in DNA2-mediated nascent strand degradation (Supplementary Fig. 4d–f). These findings indicate that K164 modification by ubiquitin, rather than SUMO, is necessary for replication fork protection.

Next, we investigated the impact of DNA2-mediated fork degradation on genomic instability. HU treatment induced 53BP1 foci preferentially in KR cells compared to WT. Importantly, DNA2 inhibition suppressed HU-induced 53BP1 foci formation in KR cells (Fig. 2e). Similar results were obtained for RPA foci (Supplementary Fig. 4g). We also measured DSB formation using the neutral comet assay. Similar to the 53BP1 foci results, this experiment indicated that HU induces DSBs at higher rates in KR cells, which depends on DNA2 activity (Fig. 2f). These findings argue that, in KR cells, DNA2-mediated processing of stalled replication forks results in DSB formation and genomic instability.

**Impact of fork reveral enzymes on fork degradation in K164R cells.** In BRCA-deficient cells, nascent strand degradation by MRE11 occurs upon fork reversal[25,26,41]. We investigated if fork reversal is also required for nascent strand degradation in KR cells. Fork reversal depends on RAD51 and the translocases HLTF, ZRANB3, and SMARCAL1[20,22,25,41–43]. Depletion of RAD51 restored nascent tract integrity (Fig. 3a; Supplementary Fig. 5a), indicating that fork reversal by RAD51 is also a prerequisite for nascent strand degradation in KR cells. We next investigated the involvement of translocases HLTF, ZRANB3, and SMARCAL1. Previously, individual depletion of each of these factors was shown to completely restore fork protection in BRCA-deficient cells, suggesting that they act in concert to perform fork reversal[25,44]. In contrast, in KR cells we observed differential impact of translocase depletion. Loss of HLTF did not restore fork protection, whereas ZRANB3 depletion partially rescued nascent strand degradation, and complete rescue was observed after depleting SMARCAL1 (Fig. 3b; Supplementary Fig. 5b). These findings demonstrate that, unlike in BRCA-mutant cells, fork reversal in PCNA-K164R cells depends on SMARCAL1 and partially on ZRANB3, but does not involve HLTF activity. It was previously shown that fork reversal by ZRANB3 depends on UBC13-catalyzed poly-ubiquitination of PCNA, which recruits

ZRANB3 to stalled forks[38]. However, ZRANB3 knockdown only partially rescued HU-induced nascent strand degradation in UBC13-depleted cells (Supplementary Fig. 5c), similar to its effect in PCNA-K164R cells. In contrast, ZRANB3 knockdown fully suppressed nascent strand degradation in BRCA2-depleted cells (Supplementary Fig. 5c), as previously shown[25,26]. These findings suggest that ZRANB3-mediated fork reversal is only partially dependent on PCNA poly-ubiquitination and it is not completely abolished in the absence of this modification.

Besides its role in fork reversal, RAD51 is also critical for the protection of reversed forks. The inability to stabilize RAD51 at stalled replication forks renders them susceptible to nucleolytic processing[23,24,28]. To test if the fork protection defect observed in KR cells is caused by defective RAD51 loading on reversed forks, we depleted RADX. RADX antagonizes RAD51 accumulation at stalled forks, and its depletion results in enhanced RAD51 binding to reversed forks[45]. While RADX knockdown suppressed nascent strand degradation in BRCA2-deficient cells, it failed to restore fork protection in KR cells (Fig. 3c; Supplementary Fig. 5d), arguing that the nascent strand degradation in KR cells is not caused by deficient RAD51-mediated fork protection.

Previously, DNA2-mediated nascent strand degradation was described in cells depleted of RECQL1, which restarts stalled replication forks upon prolonged fork arrest (treatment with 4 mM HU for 6 h)[27]. To test if the fork protection defect in KR cells is caused by defective RECQL1-mediated fork restart, we depleted RECQL1 in both WT and KR cells. Under experimental conditions used to detect nascent strand degradation in KR cells (4 mM HU for 4.5 h), we did not observe any impact of RECQL1 knockdown in either WT or KR cells (Supplementary Fig. 5e, f). These findings argue against an involvement of RECQL1 in the fork protection defect observed in KR cells.

**Okazaki fragment ligation underlies fork protection.** The ~30% increase in fork speed observed in KR cells is reminiscent of cells depleted of DNA replication factors involved in OFM such as LIG1 and FEN1[46]. Consistent with these findings, we also observed that depletion of LIG1 or FEN1 results in longer nascent tracts (Fig. 4a, b; Supplementary Fig. 6a). Thus, we investigated if perturbing OFM also results in nascent DNA degradation upon fork stalling. Under identical conditions as those used for detecting nascent strand degradation in KR cells (4 mM HU for 3 h), LIG1 or FEN1 depletion showed a similar fork protection defect (Fig. 4a, b). These findings indicate that defects in OFM result in nascent DNA degradation upon replication stress.

Using synthetic genetic array analyses in yeast, we previously uncovered a genetic similarity between the PCNA-K164R mutation and inactivation of lagging strand synthesis factors[47]. Moreover, previous work in fission yeast uncovered a lagging strand synthesis defect in PCNA-K164R cells[48]. Coupled with the fork protection defect similarities described above, these findings raise the question of whether PCNA-K164R cells have OFM defects. Previously, it was shown that LIG1 depletion in human cells results in increased poly-ADP-ribose (PAR) chain formation on chromatin, as detected upon inhibition of poly(ADP-ribose) glycohydrolase (PARG) by a specific inhibitor[49]. As increased chromatin PAR chain formation may represent a marker of OFM defects, we measured PAR chain formation in KR cells. Similarly to LIG1 depletion, PAR chain formation was enhanced in KR cells (Fig. 4c). Confirming that PAR chains are caused by accumulation of unligated OFs, their formation was suppressed upon short treatment with emetine, an inhibitor of lagging strand synthesis which prevents formation of OFs, uncoupling leading and lagging strand replication[49] (Supplementary Fig. 6b). These findings argue that LIG1-mediated Okazaki fragment ligation is

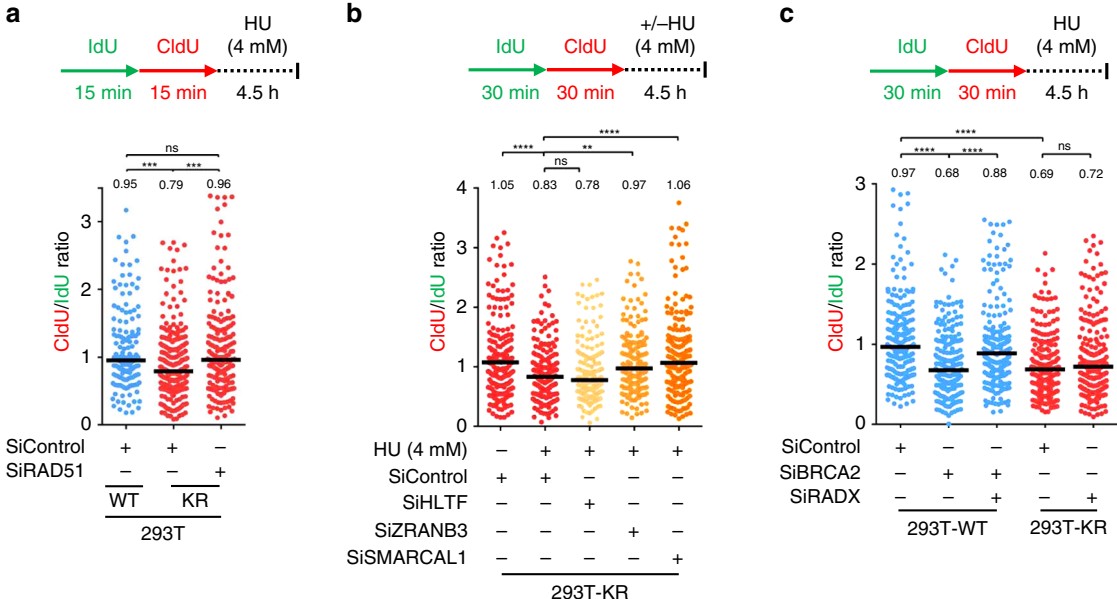

**Fig. 3 Factors involved in fork reversal are required for nascent strand degradation in PCNA-K164R cells. a** RAD51 depletion suppresses HU-induced nascent strand degradation in 293T-K164R cells. The ratio of CldU to IdU tract lengths is presented, with the median values marked on the graph and listed at the top. Asterisks indicate statistical significance (Mann–Whitney test, two-sided). A schematic representation of the fiber combing assay conditions is also presented. A western blot showing RAD51 levels upon siRNA treatment is presented in Supplementary Fig. 5a. **b** Impact of DNA translocases HLTF, ZRANB3, and SMARCAL1 on HU-induced nascent strand degradation in 293T-K164R cells. The ratio of CldU to IdU tract lengths is presented, with the median values marked on the graph and listed at the top. Asterisks indicate statistical significance (Mann–Whitney test, two-sided). A schematic representation of the DNA fiber combing assay conditions is also presented. Western blots confirming the knockdowns are shown in Supplementary Fig. 5b. **c** Knockdown of RADX suppresses HU-induced nascent strand degradation in BRCA2-deficient cells, but not in 293T-K164R cells. The ratio of CldU to IdU tract lengths is presented, with the median values marked on the graph and listed at the top. Asterisks indicate statistical significance (Mann–Whitney test, two-sided). A schematic representation of the assay conditions is also presented. Confirmation of RADX knockdown is shown in Supplementary Fig. 5d. Source data are provided as a Source Data file.

compromised in PCNA-K164R cells. Importantly, depletion of LIG1 or FEN1 did not further exacerbate nascent strand degradation, nor did it further increase fork speed, in KR cells (Fig. 4d; Supplementary Fig. 6c, d). This epistatic interaction indicates that LIG1-mediated OFM and PCNA ubiquitination may operate in the same fork protection genetic pathway.

As PCNA ubiquitination recruits non-canonical polymerases to DNA, we hypothesized that upon endogenous replication stress, the inability to recruit specialized polymerases in KR cells may result in accumulation of single stranded gaps which could hinder OF ligation on the lagging strand. To test this, we performed an alkaline comet assay on cells labeled with BrdU, allowing us to specifically detect single-stranded gaps in newly replicated DNA. KR cells showed an increase in DNA gap formation under normal replication conditions, as did RAD18-knockout cells (Fig. 4e; Supplementary Fig. 6e, f). Previously, the TLS polymerase Polκ was shown to be recruited by ubiquitinated PCNA to perform gap filling during nucleotide excision repair (NER)[50]. POLK depletion resulted in nascent strand degradation, which was partially dependent on DNA2 (Supplementary Fig. 6g, h). In contrast, depletion of the REV1 polymerase did not affect fork stability (Supplementary Fig. 6i, j). These findings show that PCNA ubiquitination suppresses accumulation of under-replicated DNA, which may otherwise interfere with OF ligation on the lagging strand.

**PCNA retention on chromatin drives fork degradation**. We next investigated how OFM defects interfere with fork stability. Previous work in yeast showed that PCNA is removed from DNA by Elg1 upon DNA Ligase I (Cdc9 in yeast)-mediated joining of OFs[51]. Indeed, knockdown of ATAD5 or LIG1 in 293T cells

resulted in increased number of PCNA chromatin foci (Fig. 5a, Supplementary Fig. 7a, b). Importantly, 293T-K164R cells also showed increased number of PCNA chromatin foci under otherwise unperturbed conditions (Fig. 5a), indicating prolonged PCNA retention on chromatin. DNA fiber combing experiments showed that, similar to LIG1 depletion, ATAD5 knockdown in WT cells results in HU-induced degradation of the nascent strand by DNA2 (Fig. 5b). However, ATAD5 knockdown in KR cells or in LIG1-depleted cells did not further exacerbate the fork protection defect, indicating that ATAD5 participates in the UbiPCNA–LIG1 genetic pathway of fork protection. These findings show that increased PCNA retention on chromatin caused by defects in OF ligation or PCNA unloading, results in nascent strand degradation upon fork arrest and reversal.

Elg1-deficient yeast cells exhibit nucleosome assembly defects, ascribed to sequestration of the PCNA-interacting CAF-1 histone chaperone in PCNA complexes on chromatin, which reduces CAF-1 availability for chromatin assembly at the replication fork[52]. In line with this, depletion of ATAD5 in human cells was shown to result in accumulation of PCNA chromatin structures which contain CAF-1 but do not actively perform DNA synthesis[4]. Thus, we investigated if nucleosome deposition is altered upon inactivation of the UbiPCNA–LIG1–ATAD5 genetic pathway. Micrococcal nuclease (MNase) sensitivity assays showed that chromatin was more accessible to nucleolytic digestion in KR, LIG1-depleted, and ATAD5-depleted cells (Fig. 5c), consistent with a reduction in nucleosome compaction[53]. This defective nucleosome packaging mirrors that observed upon depletion of the CAF-1 complex subunit CHAF1A (Fig. 5c, Supplementary Fig. 7c), suggesting that retention of PCNA on chromatin in KR cells, or upon LIG1 or ATAD5 depletion,

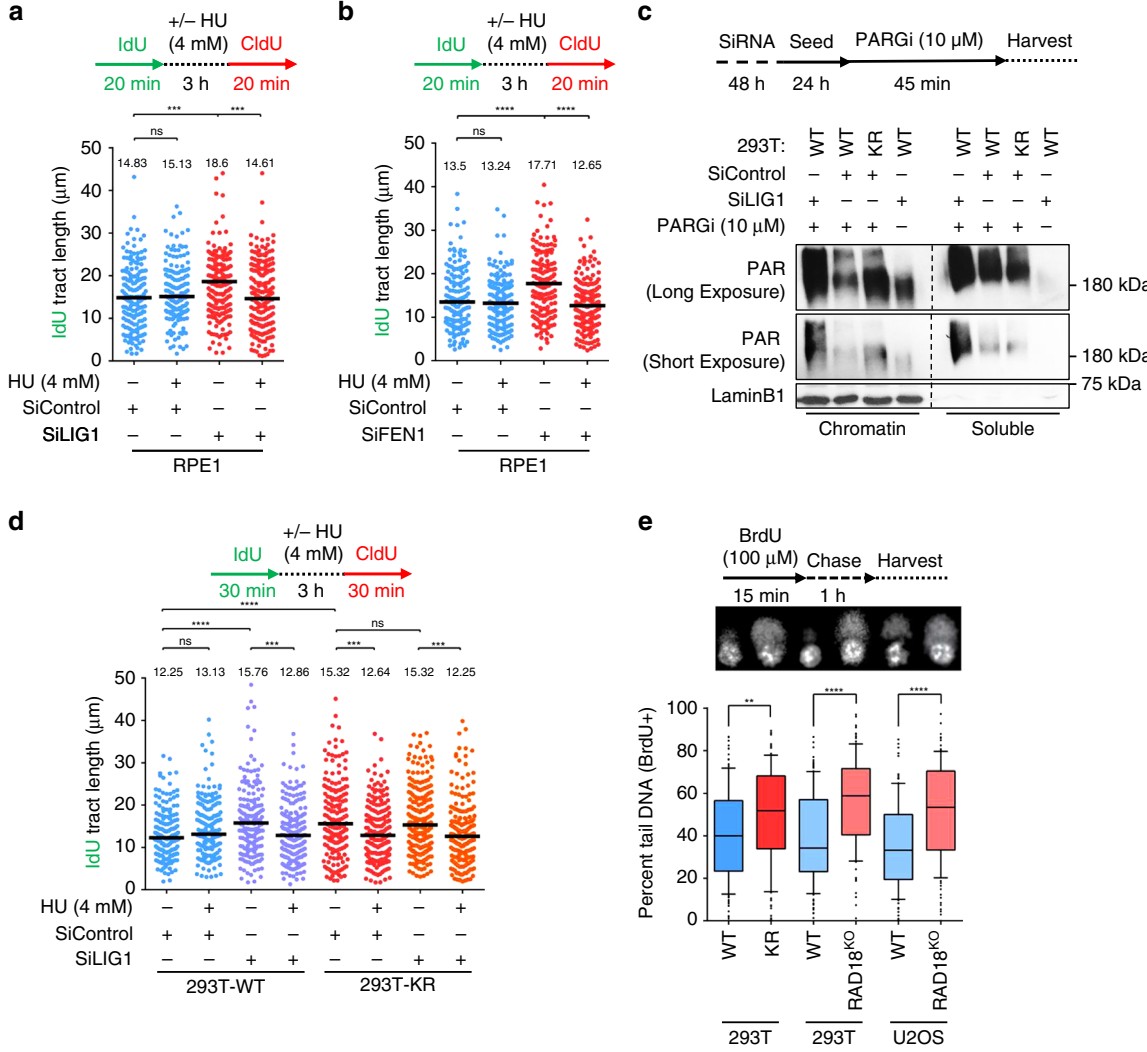

**Fig. 4 Defective Okazaki fragment maturation results in nascent strand degradation. a, b** DNA fiber combing assays showing that depletion of LIG1 **a** or FEN1 **b** results in faster replication fork progression under normal conditions, and induce nascent strand degradation upon fork arrest. The quantification of the IdU tract length is presented, with the median values marked on the graph and listed at the top. Asterisks indicate statistical significance (Mann–Whitney test, two-sided). Schematic representations of the assay conditions are also presented. Western blots confirming the knockdowns are shown in Supplementary Fig. 6a. **c** Chromatin fractionation experiment showing increased PAR chain formation in KR cells under normal growth conditions, indicative of defective Okazaki fragment maturation. Cells were treated as indicated with a PARG inhibitor (PARGi) for 45 min prior to harvesting to block PAR chain removal. LIG1 depletion was used as positive control for defective Okazaki fragment maturation. Chromatin-associated LaminB1 was used as loading control. **d** Loss of LIG1 is epistatic with the PCNA-K164R mutation for fork progression and HU-induced nascent strand degradation in 293T cells. The quantification of the IdU tract length is presented, with the median values marked on the graph and listed at the top. Asterisks indicate statistical significance (Mann–Whitney test, two-sided). A schematic representation of the assay conditions is also presented. Similar results for LIG1 and FEN1 depletion in RPE1 cells are presented in Supplementary Fig. 6c, d. **e** BrdU-alkaline comet assay showing accumulation of ssDNA gaps under normal replication conditions in 293T-K164R and RAD18-knockout 293T and U2OS cells. At least 100 cells were quantified for each condition. Center line indicates the median, bounds of box indicate the first and third quartile, and whiskers indicate the 10th and 90th percentile. Asterisks indicate statistical significance (t-test, two-tailed, unequal variance). A schematic representation of the assay conditions, and representative micrographs, are also presented. Western blots confirming RAD18 knockout, and presenting the levels of PCNA ubiquitination in these cells, are shown in Supplementary Fig. 6e, f. Source data are provided as a Source Data file.

sequesters CAF-1 away from active nucleosome deposition sites and thus interferes with its chromatin assembly function. CHAF1A depletion resulted in DNA2-mediated nascent strand degradation upon HU treatment (Fig. 5d), similar to what we previously observed upon inactivation of the Ubi PCNA–LIG1–ATAD5 genetic pathway. Altogether, these findings indicate that in KR cells, enhanced PCNA retention on chromatin results in altered nucleosome packaging likely due to aberrant CAF-1 localization (Fig. 5e; Supplementary Fig. 7d). This nucleosome packaging defect renders stalled forks

susceptible to DNA2-mediated degradation under acute replication stress (Fig. 5e; Supplementary Fig. 7e).

**Loss of PCNA ubiquitination enhances the effects of BRCA-deficiency.** In BRCA-deficient cells, fork protection correlates with resistance to cisplatin and PARPi[25,26,45]. BRCA2 depletion in KR cells enhanced nascent strand degradation upon HU treatment (Fig. 6a), indicating that they operate separately to maintain fork stability. Moreover, a synergistic increase in 53BP1 foci formation was observed in BRCA2-depleted KR cells under

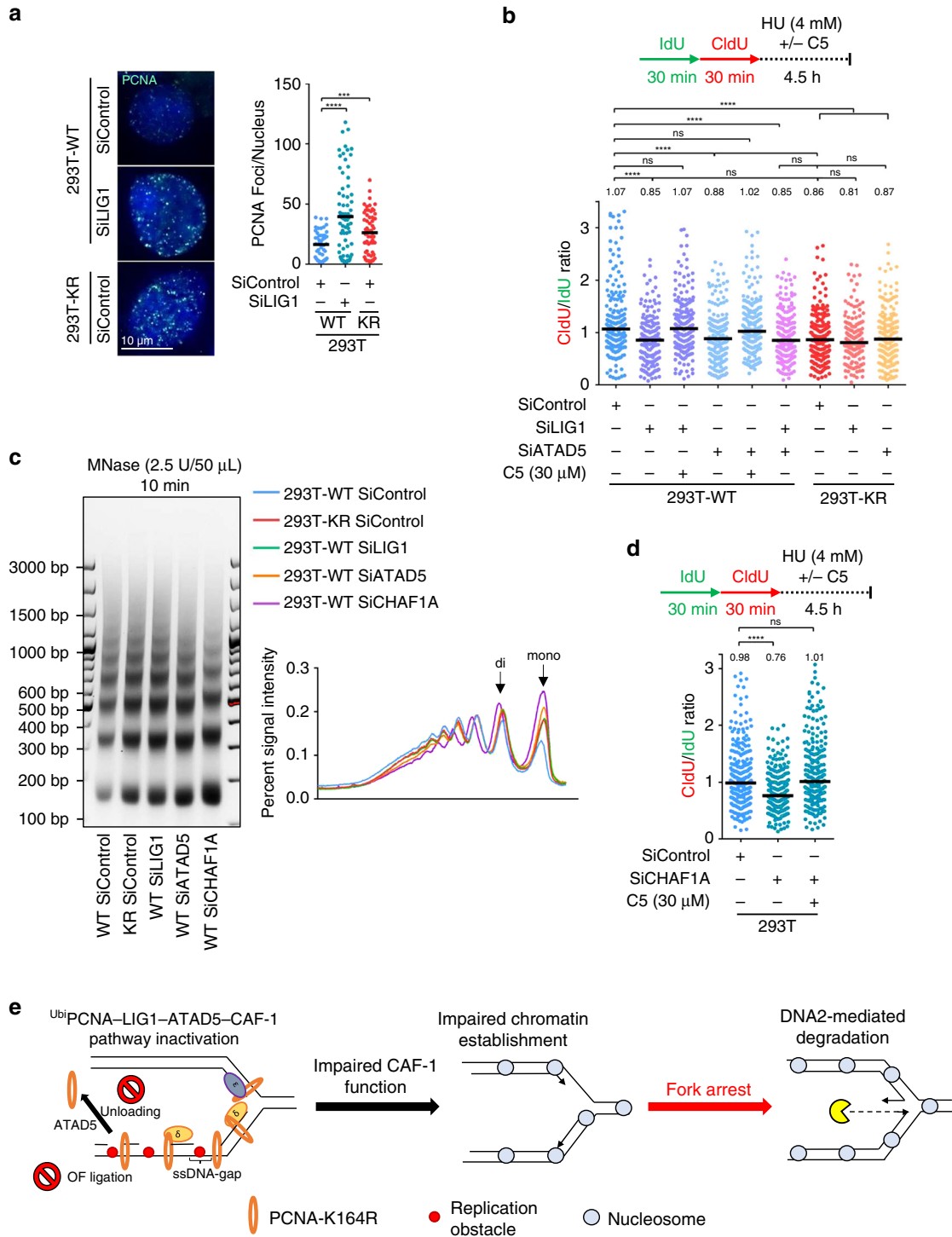

normal growth conditions (Fig. 6b). Next, we investigated if loss of BRCA2 potentiates the replication-dependent DNA damage observed in KR cells, using the BrdU alkaline comet assay. BRCA2 depletion did not affect the amount of DNA damage accumulated in newly replicated DNA during DNA synthesis (1 h chase time after BrdU pulse), in either WT or KR cells (Fig. 6c). However, 5 h after DNA synthesis, BRCA2-depleted KR cells retained a significant amount of damage in BrdU-positive DNA, compared to BRCA2-depleted WT cells or BRCA2-proficient cells (Fig. 6c). Neutral comet assays indicated that this DNA damage accumulating in BRCA2-depleted KR cells under normal growth conditions is not represented by DSBs (Supplementary

Fig. 8a). Overall, these findings indicate that ssDNA gaps which accumulate during DNA replication under normal conditions in PCNA-K164R cells, can be repaired through a BRCA2-dependent mechanism.

As replication fork stability may represent an important component of the response to PARPi in BRCA-deficient cells[25,26], we next investigated the impact of PCNA ubiquitination on olaparib sensitivity. While the KR mutation by itself did not result in olaparib sensitivity, it enhanced the sensitivity of BRCA1-depleted or BRCA2-depleted cells (Fig. 6d, e; Supplementary Fig. 8b–d). Moreover, knockdown of other components of the PCNA ubiquitination-dependent fork protection pathway

**Fig. 5 Abnormal retention of PCNA on chromatin drives replication fork degradation by altering nucleosome deposition. a** PCNA immunofluorescence showing increased PCNA retention on chromatin in 293T-K164R cells, or upon LIG1 depletion. At least 65 cells were quantified for each condition. The mean values are marked on the graph, and asterisks indicate statistical significance (t-test, two-tailed, unequal variance). Representative micrographs are also shown. **b** ATAD5 knockdown results in DNA2-mediated nascent strand degradation upon HU-induced replication fork arrest, which is epistatic to the K164R mutation. The ratio of CldU to IdU tract lengths is presented, with the median values marked on the graph and listed at the top. Asterisks indicate statistical significance (Mann–Whitney test, two-sided). Confirmation of ATAD5 knockdown is shown in Supplementary Fig. 7b. **c** Micrococcal nuclease sensitivity assay showing altered nucleosome deposition in 293T-K164R cells, as well as upon depletion of LIG1, ATAD5, or CHAF1A. A quantification of the signal intensity is also shown. Confirmation of CHAF1A knockdown is shown in Supplementary Fig. 7c. **d** CHAF1A depletion results in HU-induced degradation of nascent DNA by DNA2 nuclease. The ratio of CldU to IdU tract lengths is presented, with the median values marked on the graph and listed at the top. Asterisks indicate statistical significance (Mann–Whitney test, two-sided). **e** Schematic representation of the proposed model depicting the UbiPCNA–LIG1–ATAD5–CAF-1 genetic pathway. Under unperturbed growth conditions, PCNA ubiquitination is required for efficient lagging strand synthesis by mediating gap-filling behind progressing replication forks. In the absence of PCNA ubiquitination, persistent gaps are formed between the lesion and the previous OF. These gaps interfere with OF maturation and subsequent PCNA unloading by ATAD5. By sequestering the CAF-1 chromatin assembly complex, PCNA retention on the lagging strand alters the efficiency of chromatin establishment which results in replication forks encountering a sparse chromatin organization. Upon fork arrest and reversal, the abnormal structure generated is a substrate for uncontrolled resection by DNA2. Source data are provided as a Source Data file.

described here, namely RAD18 and ATAD5, also enhanced the olaparib sensitivity of BRCA2-knockout HeLa cells (Fig. 6f, g). These findings show that PCNA ubiquitination provides an alternative mechanism for fork protection and replication stress suppression in BRCA-deficient cells, and determines PARPi sensitivity in these cells.

## Discussion

Our work uncovers a mechanism of replication fork protection controlled by PCNA ubiquitination (Fig. 5e; Supplementary Fig. 7d, e). We show that PCNA ubiquitination promotes gap filling during normal S-phase. In the absence of PCNA ubiquitination, OF ligation is perturbed, likely because of accumulated gaps on the lagging strand. OF ligation, in turn, permits unloading of PCNA by ATAD5, which enables accurate chromatin assembly by CAF-1. Importantly, we show that defective OF ligation, and the subsequent chromatin assembly deficiency, render cells susceptible to DNA2-mediated nascent strand degradation upon fork stalling and reversal, resulting in double strand break formation and genomic instability.

Our previous work using synthetic genetic array analyses in yeast showed a striking genetic similarity between the PCNA-K164R mutation and inactivation of FEN1 (Rad27 in yeast) and other lagging strand synthesis factors, implicating PCNA ubiquitination in OFM[47]. Moreover, we showed that PCNA ubiquitination was increased in RAD27 and CDC9 mutants, and identified a synthetic lethal interaction between the PCNA-K164R mutation and RAD27 inactivation[54–56]—further indicating that PCNA ubiquitination is required for OF processing. We show here that human PCNA-K164R cells exhibit hallmarks of lagging strand synthesis defects, and accumulate ssDNA which may directly interfere with OF ligation by LIG1. In yeast, PCNA is preferentially enriched on the lagging strand during normal DNA replication[57]. It is thus likely that the single-stranded gaps observed in KR cells are on the lagging strand. Indeed, as the lagging strand experiences frequent repriming due to the discontinuous mode of DNA replication, Polδ arrest at endogenous sites of replication stress would result in accumulation of gaps behind the fork as a new DNA synthesis reaction is initiated upon regular repriming of the subsequent OF[58,59]. In contrast, stalling of Polε on the leading strand requires a dedicated repriming event which needs to be quickly put in place to resume replication, thus accumulation of gaps on this strand is less likely. While engagement of TLS polymerases requires PCNA mono-ubiquitination, we found that the ubiquitin ligase UBC13, involved in PCNA polyubiquitination, also protects against DNA2-mediated nascent strand degradation. A role for UBC13 in TLS has been

previously proposed[60], perhaps explaining these findings. Alternatively, it is possible that polyubiquitinated PCNA-dependent template switching also participates in gap filling during OFM, or that other substrates of UBC13 are involved. Finally, it has been previously proposed that in fission yeast, PCNA ubiquitination enhances its interaction with Polδ[48], suggesting that the activity of the lagging strand replicative polymerase itself may be defective in KR cells.

We demonstrate here that inactivation of the UbiPCNA–LIG1–ATAD5–CAF-1 genetic pathway results in DNA2-mediated nascent strand degradation upon fork reversal. DNA2 was previously shown to degrade stalled forks upon prolonged replication stress in wildtype cells[27], but in KR cells degradation occurs upon much shorter HU exposure, which does not affect fork stability in wildtype cells. This indicates that PCNA ubiquitination specifically suppresses DNA2-mediated processing of stalled forks. Our results indicate that this process is mechanistically different than the nascent strand degradation described in BRCA-deficient cells, which involves the activity of MRE11 on reversed forks unprotected by RAD51[23–26]. In PCNA-K164R cells, the BRCA pathway is intact and the fork protection defect in these cells is likely not caused by defective RAD51 loading. Instead, we propose that the aberrant nucleosome deposition promotes nascent strand degradation by DNA2. DNA2 is also involved in fork degradation in BRCA-deficient cells[26], but that activity is performed in conjunction with MRE11 and thus is different than what we report here in PCNA-K164R cells, where we find no evidence of MRE11 activity.

In BRCA-deficient cells, depletion of any of the three translocases ZRANB3, HLTF, and SMARCAL1 completely restored fork protection, indicating that in this genetic context, they work in concert to perform fork reversal[25,44]. In contrast, in KR cells the three translocases have differential impacts, suggesting that they do not necessarily have to act together in fork reversal. HLTF depletion did not have any effect on fork degradation in KR cells. Besides its translocase activity, HLTF contains a RING ubiquitin ligase domain which catalyzes K63-linked poly-ubiquitination of PCNA at K164, building upon the single ubiquitin moiety initially added by RAD18[32,33]. Our findings suggest that HLTF needs to ubiquitinate PCNA in order to perform its translocase activity. ZRANB3 had a moderate impact on fork protection. Although ZRANB3 preferentially binds poly-ubiquitinated PCNA, it also interacts with unmodified PCNA through its PIP-box[61], thereby explaining its intermediate phenotype. Indeed, ZRANB3 knockdown could partially suppress HU-induced fork degradation in UBC13-depleted cells, indicating that its activity is not fully dependent on PCNA poly-ubiquitination. Lastly, depletion of

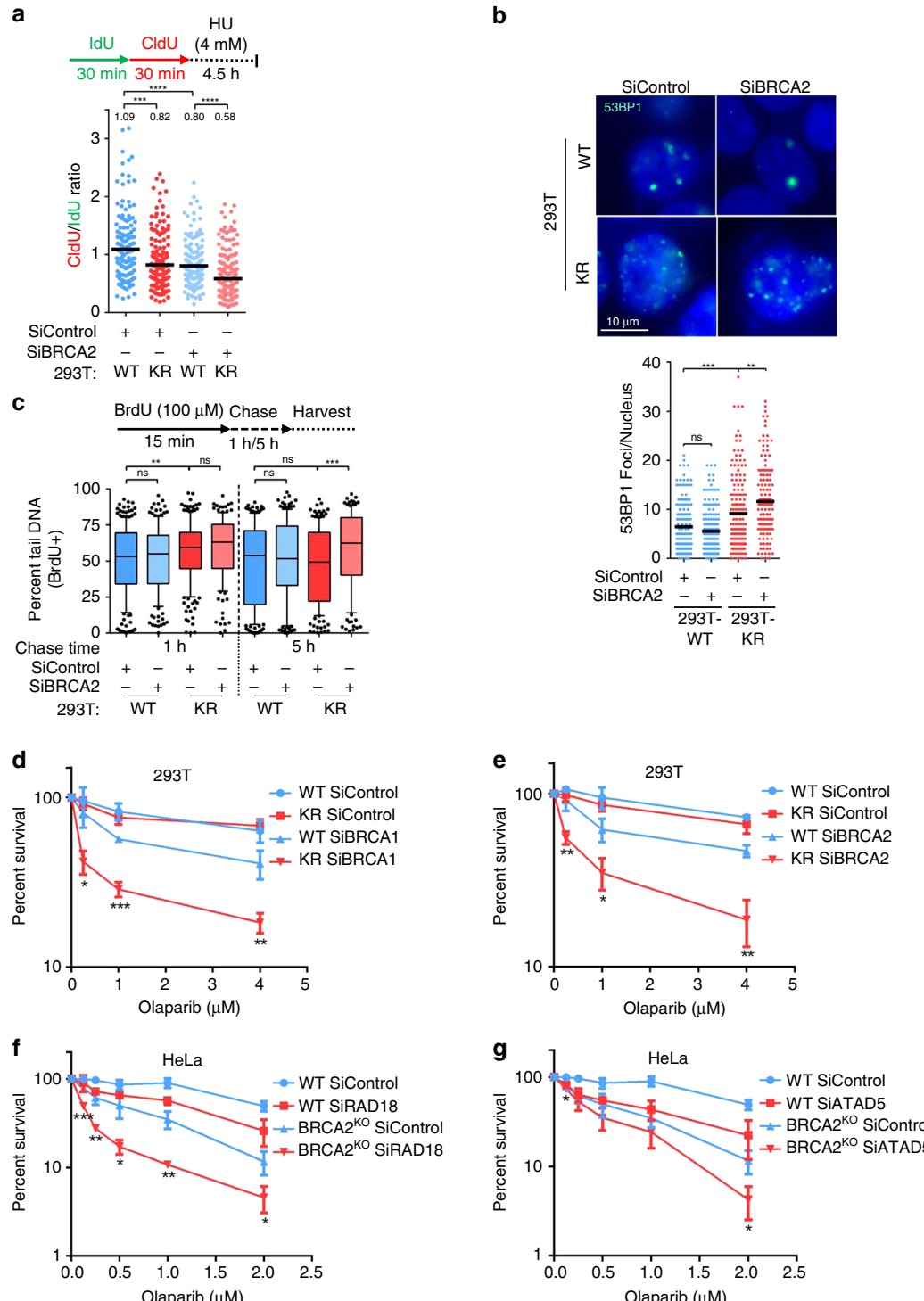

SMARCAL1 completely suppressed fork degradation, indicating that SMARCAL1 is the primary fork reversal activity operating in KR cells. Previous work showed that fork reversal is defective in mouse K164R cells[38]; our findings that depletion of RAD51, SMARCAL1, and ZRANB3 can suppress nascent strand degradation suggest that fork reversal in PCNA-K164R cells is impaired, but not abolished. We hypothesize that this residual fork reversal unveils the substrate for DNA2-mediated degradation. However, it is also possible that DNA2-mediated nascent DNA nucleolysis does not occur on reversed forks, but on some other type of structures formed at stalled replication forks by SMARCAL1 activity in PCNA-K164R cells.

In order for nascent strand degradation to be detectable by the DNA fiber combing assay, both nascent strands in symmetrically reversed forks must be degraded. During DSB resection, in order to generate the 3′ overhangs, MRE11 initiates a nick on the complementary strand, followed by resection in a 3′–5′ manner towards the DSB, thereby allowing access to long-range 5′–3′ nucleases[62,63]. It is thus conceivable that in BRCA-deficient cells that lack RAD51-mediated protection of reversed forks, MRE11 is able to attack both nascent strands using its 3′–5′ nuclease activity. In contrast, DNA2 can only perform resection in the 5′–3′ direction. Aside from DSB resection, DNA2 is also capable of processing long 5′ss-DNA flaps arising from excessive strand

**Fig. 6 Genetic interaction between PCNA ubiquitination and the BRCA pathway. a** DNA fiber combing assay showing that concomitant loss of BRCA2 and PCNA ubiquitination enhances nascent strand degradation upon HU-induced fork arrest. The ratio of CldU to IdU tract lengths is presented, with the median values marked on the graph and listed at the top. Asterisks indicate statistical significance (Mann–Whitney test, two-sided). A schematic representation of the fiber combing assay conditions is also presented. **b** Immunofluorescence experiment showing synergistic increase in 53BP1 foci formation in unsynchronized K164R cells upon BRCA2 depletion. At least 100 cells were quantified for each condition (pooled from two experiments). The mean values are marked on the graph, and asterisks indicate statistical significance (*t*-test, two-tailed, unequal variance). Representative micrographs are also shown. **c** BrdU-alkaline comet assay measuring ssDNA gaps under normal replication conditions in BRCA2-depleted 293T cells at 1 and 5 h after BrdU pulse. At least 100 cells were quantified for each condition. Center line indicates the median, bounds of box indicate the first and third quartile, and whiskers indicate the 10th and 90th percentile. Asterisks indicate statistical significance (*t*-test, two-tailed, unequal variance). A schematic representation of the assay conditions is also presented. **d**, **e** Clonogenic survival experiments showing that loss of PCNA ubiquitination does not result in olaparib sensitivity, but it drastically increases the olaparib sensitivity of BRCA1-depleted **d** and BRCA2-depleted **e** cells. The average of three experiments, with standard deviations indicated as error bars, is shown. Asterisks indicate statistical significance (*t*-test, two-tailed, unequal variance) comparing WT siBRCA1/2 to KR siBRCA1/2. Representative images of the clonogenic assays are presented in Supplementary Fig. 8c, d. **f**, **g** Clonogenic survival experiments showing that depletion of RAD18 **f** or ATAD5 **g** increases the olaparib sensitivity of BRCA2-knockout HeLa cells. The average of three experiments, with standard deviations indicated as error bars, is shown. Asterisks indicate statistical significance (*t*-test, two-tailed, unequal variance) comparing BRCA2$^{KO}$ siControl to BRCA2$^{KO}$ siRAD18/siATAD5. Source data are provided as a Source Data file.

displacement during OF synthesis[64–66]. Thus, we speculate that the substrate for DNA2 in PCNA-K164R cells is represented by reversed forks with 5′-overhangs arising from the lagging strand, which would resemble 5′ss-DNA flaps formed during OFM. How might the asymmetry between leading and lagging strands in PCNA-K164R arise? OFM defects in these cells result in retention of PCNA on chromatin, which sequesters CAF-1 resulting in aberrant chromatin assembly[52]. Nucleosome positioning can alter OF periodicity, and disrupting chromatin assembly by inactivating CAF-1 results in abnormally long OFs[67,68]. Moreover, nucleosome positioning guides the priming activity of Polα-primase[69]. Therefore, we speculate that in KR cells, replication forks encounter sparse chromatin organization due to impaired CAF-1 activity, which alters the periodicity of OF priming resulting in longer OFs since they initiate further ahead of leading strand synthesis. Upon fork arrest and reversal, the longer OFs would give rise to 5′-overhangs (Fig. 5e; Supplementary Fig. 7d, e).

Replication fork protection is considered an important component of PARPi resistance in BRCA-deficient cells[25,26]. We show here that concomitant loss of PCNA ubiquitination and the BRCA pathway results in increased fork degradation, and synergistically enhances DNA damage accumulation and olaparib sensitivity. These findings suggest that PCNA ubiquitination provides a survival mechanism for BRCA-deficient cells exposed to DNA damaging agents. While nascent strand degradation is associated with sensitivity of BRCA-deficient cells to cisplatin and PARPi, the exact contribution of fork protection to PARPi resistance is unclear[25,26,45]. Our findings that PCNA-K164R cells have fork protection defects but no olaparib sensitivity argue against a role for fork protection in PARPi resistance. Instead, our work suggests that the synthetic lethality between BRCA-deficiency and PARPi[70,71] may in part reflect perturbations in OFM. Recent studies revealed a previously unknown role of PARP1 as a sensor of unligated OFs during unperturbed S-phase[49]. Therefore, it is possible that a major effect of PARPi is preventing the ligation of OFs that have failed conventional ligation by LIG1. Yet it has remained unclear whether perturbed OF ligation causes toxicity in BRCA-deficient cells. In this study, we show that cells with intact BRCA1/2 function are able to tolerate unligated OFs resulting from the accumulation of ssDNA gaps generated in the absence of PCNA ubiquitination. In contrast, BRCA-deficient cells depend on PCNA ubiquitination to suppress accumulation of ssDNA gaps under normal growth conditions. In BRCA-mutant cells, upon perturbation of OF ligation by PARPi treatment, the concomitant loss of PCNA ubiquitination becomes toxic. These observations point towards perturbation of OF ligation as a potential mechanism underlying the synthetic lethality of PARPi in cells with BRCA deficiency and reflect the importance of PCNA ubiquitination in mediating tolerance to PARPi in these cells.

## Methods

**Cell culture and protein techniques**. Human 293T, RPE1, HeLa, and U2OS cells were grown in DMEM supplemented with 10% fetal calf serum.

To generate the K164R cells, the gRNA sequences used were: TTTCACTCCGT CTTTTGCACAGG for 293T cells and GCAAGTGGAGAACTTGGAAATGG for RPE1 cells. The sequences were cloned into the pX458 vector (pSpCas9BB-2A-GFP; obtained from Addgene). Cells were co-transfected with this vector and a repair template spanning the K164 genomic locus but containing the K164R mutation (AAA-AGA codon change). Transfected cells were FACS-sorted into 96-well plates using a BD FACSAria II instrument. Resulting monoclonal cultures were screened by western blot for loss of PCNA ubiquitination using an antibody specific for this modification. For verification of positive cell lines, the targeted genomic region was PCR amplified from genomic DNA, cloned into pBluescript, and multiple clones were Sanger-sequenced to ensure that all alleles are identified. In the 293T KR5 clone obtained, no wildtype allele was discovered, and at least one K164R allele was found. In addition, several other alleles bearing insertions or deletions at that position, introducing frameshifts and premature stop codons, were identified. No truncated forms were detectable by western blot. In RPE cells, only alleles with the K164R mutation were detected. For exogenous PCNA expression, pLV-puro-CMV lentiviral constructs encoding wildtype or the K164R variant were obtained from Cyagen. Infected cells were selected with puromycin.

For RAD18 gene knockout, the commercially available CRISPR/Cas9 KO Plasmid BRCA2 CRISPR/Cas9 KO plasmid was used (Santa Cruz Biotechnology sc-406099). Transfected cells were FACS-sorted into 96-well plates using a BD FACSAria II instrument. Resulting colonies were screened by western blot. The BRCA2-knockout HeLa cells were previously published[72].

Denatured whole cell extracts were prepared by boiling cells in 100 mM Tris, 4% SDS, and 0.5 M β-mercaptoethanol. Chromatin fractionation was performed by subjecting cells to extraction with 0.1% Triton-X. For PARG inhibition, cells were incubated with 10 µM PDD00017273 (Sigma SML1781) for 45 min prior to harvesting. Emetine was obtained from Sigma-Aldrich, cisplatin was obtained from Biovision, while olaparib and mirin were obtained from Selleck Chemicals.

Antibodies used for western blot were: PCNA (Cell Signaling Technology 2586); Ubiquityl-PCNA Lys164 (Cell Signaling Technology 13439); CHK2 (Cell Signaling Technology 2662); pCHK2-T68 (Cell Signaling Technology 2661); CHK1 (Cell Signaling Technology 2360); pCHK1-S317 (Cell Signaling Technology 2344); BRCA2 (Bethyl A303-434A); EXO1 (Santa Cruz Biotechnology sc-56092); WRN (Santa Cruz Biotechnology sc-5629); CTIP (Santa Cruz Biotechnology sc-271339); MUS81 (Santa Cruz Biotechnology sc-47692); RAD18 (Cell Signaling Technology 9040); UBC9 (Santa Cruz Biotechnology sc-10759); RAD51 (Santa Cruz Biotechnology sc-8349); ZRANB3 (Bethyl A303-033A); HLTF (Santa Cruz Biotechnology sc-398357); SMARCAL1 (Santa Cruz Biotechnology sc-376377); RECQL1 (Santa Cruz Biotechnology sc-166388); PAR chains (ENZO ALX-804-220); LaminB1 (Abcam ab16048); LIG1 (Santa Cruz Biotechnology sc-271678); REV1 (Santa Cruz Biotechnology sc-393022); POLK (Santa Cruz Biotechnology sc-166667); CHAF1A (Cell Signaling Technology 5480); BRCA1 (Santa Cruz Biotechnology sc-642); Vinculin (Santa Cruz Biotechnology sc-73614); GAPDH (Santa Cruz Biotechnology sc-47724); FEN1 (Santa Cruz Biotechnology sc-28355); UBC13 (Santa Cruz Biotechnology sc-376470). All antibodies were used at a dilution of 1:500. Uncropped scans of all blots are provided as a Source Data file.

For gene knockdown, cells were transfected with Stealth siRNA (Life Tech) using Lipofectamine RNAiMAX reagent. AllStars Negative Control siRNA (Qiagen 1027281) was used as control. Gene knockdown was confirmed by western blot or qRT-PCR, and representative results are shown. The siRNA-targeting sequences used were: BRCA2: GAGAGGCCTGTAAAGACCTTGAATT; EXO1: CCTGTTG AGTCAGTATTCTCTTTCA; WRN: TGGGCTCCTGCAGACATTAACTTAA; C TIP: GGGTCTGAAGTGAACAAGATCATTA; MUS81: TTTGCTGGGGTCTCT AGGATTGGTCT; RAD18: CATATTAGATGAACTGGTATT; UBC9: TAAACA AGCCTCCTTCCCACGGAGT; RAD51: CCATACTGTGGAGGCTGTTGCCTA T; HLTF: TGCATGTGCATTAACTTCATCTGTT; ZRANB3: TGGCAATGTAGT CTCTGCACCTATA; SMARCAL1: CACCCTTTGCTAACCCAACTCATAA; RECQL1: ACAGGAGGUGGAAAGAGCTTATGTT; REV1: GAAATCCTTGCA GAGACCAAACTTA; POLK: CAGCCATGCCAGGATTTATTGCTAA; ATAD5: GGTACGCTTTAAGACAGTTACTGTT; BRCA1: AATGAGTCCAGTTTCGT TGCCTCTG; LIG1: Silencer Select ID s8174 (Life Technologies); FEN1: Silencer Select ID s5104 (Life Technologies); CHAF1A: Silencer Select ID s19499 (Life Technologies); UBC13 Silencer Select ID: s14595 (Life Technologies); RADX: ON-TARGETplus J-014634-21 (Dharmacon).

**Functional assays**. For clonogenic experiments, 1000–2000 cells (depending on plating efficiency) were seeded in six-well plates. For UV sensitivity, cells were treated 24 h after seeding. For cisplatin and olaparib treatment, cells were seeded in indicated drug concentrations for 24 and 72 h, respectively, followed by media change. Two weeks later, colonies were stained with Crystal violet. For time-course proliferation experiments, 500 cells were seeded in wells of 96-well plates, and cellular viability was scored at indicated days using the CellTiterGlo reagent (Promega G7572). EdU incorporation was assayed using the Click-iT Plus kit (Invitrogen C10633) according to the manufacturer's instructions, using a BD FACSCanto II flow cytometer, and analyzed using FACSDiva 8.0.1 software.

For the SupF assay[36], cells were transfected with UVC-irradiated (1000 J/m$^2$) pSP189 (SupF) plasmid. Three days later, the plasmid was recovered using a miniprep kit (Promega), DpnI digested and transformed into MBM7070 indicator bacteria. Transformants were selected on plates containing 1 mM IPTG and 100 μg/ml X-gal. The ratio of white (mutant) to total (blue + white) colonies was scored as mutation frequency.

**Comet assay**. For the BrdU alkaline comet assay, cells were incubated with 100 μM BrdU for 15 min, followed by media removal, PBS wash, and incubation in fresh media for 1 h. Cells were harvested and subjected to the alkaline comet assay using the CometAssay kit (Trevigen 4250-050) according to the manufacturer's instructions. Slides were stained with primary anti-BrdU (BD 347580) and secondary Alexa Fluor 568 (Invitrogen A11031) antibodies. Slides were mounted with DAPI-containing Vectashield mounting medium (Vector Labs) and imaged using a Nikon Eclipse TE2000-U microscope. The percent tail DNA was calculated using CometScore 2.0 software. For the neutral comet assay, cells were treated as indicated, harvested, and the assay was performed using the CometAssay kit (Trevigen 4250-050) according to the manufacturer's instructions. Slides were mounted, imaged, and analyzed as described above.

**DNA fiber assay**. Cells were incubated consecutively with 100 μM CldU and 100 μM IdU for the indicated times. HU and nuclease inhibitors (50 μM mirin for MRE11 inhibition or 30 μM C5[73] for DNA2 inhibition) were added as indicated. Next, cells were harvested and DNA fibers were obtained using the FiberPrep kit (Genomic Vision EXT-001). DNA fibers were stretched on glass coverslips (Genomic Vision COV-002-RUO) using the FiberComb Molecular Combing instrument (Genomic Vision MCS-001). Slides were incubated with primary antibodies (Abcam 6326 for detection of CldU; BD 347580 for detection of IdU; Millipore Sigma MAB3034 for detection of DNA), washed with PBS, and incubated with Cy3, Cy5, or BV480-coupled secondary antibodies (Abcam 6946, Abcam 6565, BD Biosciences 564879). Following mounting, slides were imaged using a Leica SP5 confocal microscope and analyzed using LASX 3.3.0.16799 software. At least 100 replication tracts were quantified for each sample.

**Immunofluorescence**. Cells were seeded on sterile glass coverslips coated with poly-L-lysine (Sigma P8920) as per manufacturer's instructions and allowed to incubate for 24 h. For RPA and 53BP1 foci detection, cells were fixed with 3.7% paraformaldehyde for 15 min, followed by three washes with PBS. Cells were then permeabilized with 0.5% Triton X-100 for 10 min. After two washes with PBS, slides were blocked with 3% BSA in PBS for 15 min, followed by incubation with the primary antibody diluted in 3% BSA in PBS, for 2 h at room temperature. After three washes with PBS, the secondary antibody was added for 1 h. Slides were mounted with DAPI-containing Vectashield mounting medium (Vector Labs). For PCNA foci detection, cells were pre-extracted with 0.5% Triton X-100, followed by one PBS wash and methanol fixation for 30 min at −20 °C. After two washes with PBS, cells were blocked with 3% BSA in PBS for 15 min. Primary and secondary antibody treatments as well as mounting were performed as mentioned above. Primary antibodies used for immunofluorescence were: 53BP1 (Bethyl A300-272A); RPA32 (Abcam ab2175); PCNA (Cell Signaling Technology 2586). Secondary antibodies used were AlexaFluor 488 or AlexaFluor 568 (Invitrogen

A11001, A11008, A11031, and A11036). Slides were imaged using a DeltaVision Elite confocal microscope. The number of foci/nucleus was quantified using ImageJ 1.52p software. For the PCNA immunofluorescence, in order to remove non-S-phase cells, only cells with at least two foci were quantified, corresponding to the top three quartiles in wildtype, were included in the quantification.

**Quantification of gene expression by real-time quantitative PCR (RT-qPCR)**. Total mRNA was purified using TRIzol reagent (Life Tech). To generate cDNA, 1 μg RNA was subjected to reverse transcription using the RevertAid Reverse Transcriptase Kit (Thermo Fisher Scientific) with oligo-dT primers. Real-time qPCR was performed with PerfeCTa SYBR Green SuperMix (Quanta), using a CFX Connect Real-Time Cycler (BioRad). The cDNA of GAPDH gene was used for normalization. Primers used were: RADX for: ATGATGTGACGATCTCAGA TGGG; RADX rev: CCCCTGGCCTATCCTTTTCTC; ATAD5 for: AGGAAG AGATCCAACCAACG; ATAD5 rev: ATGTTTCGAAGGGTTGGCAG; GAPDH for: AAATCAAGTGGGGCGATGCTG; GAPDH rev: GCAGAGATGATGACCCTTTTG.

**MNase assay**. Cells were trypsinized, washed with PBS and lysed with cold NP-40 lysis buffer (10 mM Tris–HCl, 10 mM NaCl, 3 mM MgCl$_2$, 0.5% NP-40, 0.15 mM spermine, and 0.5 mM spermidine) for 5 min. The resulting nuclei were washed once and resuspended in MNase digestion buffer (10 mM Tris–HCl, 15 mM NaCl, 60 mM KCl, 1 mM CaCl$_2$, 0.15 mM spermine, and 0.5 mM spermidine). Resuspended cells were digested with 2.5 U MNase in digestion buffer for the indicated times. The reaction was stopped by adding an equal volume of the MNase stop buffer (100 mM EDTA, 10 mM EGTA, 15 mM NaCl, 60 mM KCl, and 2% SDS) followed by proteinase K digestion (final concentration of 0.0375 μg/μl) at 37 °C overnight. DNA was isolated using phenol–chloroform extraction and subsequently subjected to RNAse A digestion. Samples were run on 1.6% agarose gels and visualized using GelRed. Signal intensities were quantified using Icy 2.0.2.0 software.

**Statistics and reproducibility**. For the DNA fiber experiments, the Mann–Whitney statistical test was performed. For all other assays, the statistical analysis performed was the $t$-test (two-tailed, unequal variance). Statistical significance is indicated for each graph (ns = not significant, for $p > 0.05$; * for $p < 0.05$; ** for $p < 0.01$; *** for $p < 0.001$; **** for $p < 0.0001$). The exact $p$-values are listed for each figure in the Source Data file. Primary data was recorded in Microsoft Excel 16 and statistical analyses were performed using GraphPad Prism 6 software. Western blot experiments were reproduced at least three times. DNA fiber assays, immunofluorescence, and comet assays were reproduced at least two times.

**Reporting summary**. Further information on research design is available in the Nature Research Reporting Summary linked to this article.

## Data availability

Data supporting the findings of this manuscript are available from the corresponding author upon reasonable request. The source data underlying Figs. 1a–h, 2a–f, 3a–c, 4a–e, 5a–d, 6a–g, and Supplementary Figs. 1a–e, 2a–s, 3a–f, 4a–j, 4a–c, 6a, b are provided as a Source Data file.

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

## Acknowledgements

We would like to thank Drs. Alessandro Vindigni, David Cortez, Alberto Ciccia, James Broach, and Mark Hedglin for materials and advice; and the Penn State College of Medicine Flow Cytometry and Imaging cores. This work was supported by: NIH R01GM134681 (to A.-K.B. and G.-L.M.); NIH R01ES026184 (to G.-L.M.); NIH R01GM074917 and NIH T32-CA009138 (to A.-K.B.); NIH R01CA073764 (to B.S.).

## Author contributions

T.T., W.L., C.M.N., and K.E.C. conducted the experiments; B.S. provided the C5 compound; T.T., A.-K.B., and G.-L.M. designed the experiments and wrote the paper.

## Competing interests

The authors declare no competing interests.
