## [Peer Review File · Nature Communications]

Reviewers' comments:

Reviewer #1 (Remarks to the Author):

In this study, the authors have described how PCNA mono-ubiquitination protects stalled replication forks from DNA2-mediated degradation and its role in promoting gap-filling during normal S-phase. The authors have performed their studies in 293T and RPE cells that express PCNA that is not mono-ubiquitinated due to a lysine to arginine substitution at residue 164. These cells have reduced proliferate rate, have longer DNA fiber track length and have more DNA damage under normal growth conditions. The authors explored the cause of replicative stress in these cells under normal conditions and found that increased retention of PCNA on the chromatin resulted in defects in Okazaki fragment ligation as well PCNA unloading. Loss of Ligase 1 involved in Okazaki fragment maturation and ATAF5, which is involved in PCNA unloading also resulted in a phenotype similar to the KR mutant cells. Persistence of PCNA on the chromatin in these cells led to reduced chromatin compaction due aberrant CAF1 localization, making them more accessible to nucleolytic digestion.

Overall, the findings are very interesting and reveal a novel role for PCNA in maintenance of genomic integrity via its role in replication fork protection. The authors have gone into great detail to convincingly show the impact of PCNA mono-ubiquitination on maturation of Okazaki fragments and chromatin assembly. They have uncovered that disruption of the UbiPCNA-LIG1-ATAD5-CAF1 pathway contributes to DNA2-mediated degradation of nascent strand at stalled forks. In spite of these strengths of the study, there are inconsistencies in the way the experiments are performed to examine the impact on fork degradation. The authors have switched between using fiber track length and ratio of IdU and CldU to measure fork degradation (For example, in Figure 3A and 3B CldU/IdU length is measured but in figure 3C IdU track length is measured). While both methods are correct, the authors should have used the same method in both cell lines throughout the study for consistency. Another concern is that the effects on fork protection are very subtle. Given the subtle effects of KR mutation on the ratio of DNA fiber track lengths, it is even more important to perform all studies in a similar way and under identical conditions.

Some of the minor concerns are:

1. Figure 2C, D: C5 and Mirin treatment in WT cells is missing
2. Figure S2F: Why is the CldU track length significantly reduced in HU treated 293T-WT cells compared to untreated cells? There should be no significant change (as shown in Fig S2D in RPE cells).
3. Figure S2Q and R and Figure 4D: What is level of Ub-PCNA in RAD18 KD and KO cells?
4. Figure S2S: P values are switched.
5. Figure S2T, RPA foci are not visible. Signal is very diffused.
6. Figure S4G: No effect of REV1 KD in WT RPE cells is surprising. Since REV1 is known to promote PCNA mono-ubiquitination, which is dependent on RAD18 (Wang et al, 2016, J Cell Sc.), why is the IdU track length reduced in the absence of HU. Based on the impact of K164R mutation on track length in normal cells (Fig 2B), the IdU track length should have increased in REV1 KD cells. What is the level of mUb-PCNA in the REV1KD cells.
7. Figure 4C: Average IdU track lengths are missing in this panel. The impact of KR mutation on IdU track length (in siControl, no HU) here is very modest compared to the effects seen in Figure 1A.
8. Figure 6C: no error bar for WT siBRCA1 at 1 μ M olaparib. Why do 293T-KR cells exhibit no sensitivity to olaparib. Yet, when combined with BRCA1/2 loss, KR cells exhibit a significant increase in sensitivity compared to BRCA1/2 loss alone.

Reviewer #2 (Remarks to the Author):

Many laboratories now study nascent DNA degradation upon replication stress. It has been observed that in the presence of replication stress, replication forks generally slow down, which is accompanied by fork reversal. Under pathological conditions such as in the absence of BRCA1/2, or prolonged stalling, DNA at reversed forks gets degraded by either (or a combination of) MRE11, DNA2 and EXO1 nucleases. However, the interpretation of these experiments is often complicated, as e.g. the RAD51 protein is involved both in fork reversal and fork protection. Therefore, the RAD51 functions affect fork stability in opposing ways. When assaying nascent DNA at replication forks by fiber assays, the observed effects are the net result of both reversal and protection, and i.e. it is impossible to directly distinguish between fork slowdown and enhanced degradation. Understanding the underlying mechanisms is of great interest to the field and beyond.

In this manuscript, Thakar et al investigate the function of PCNA ubiquitination, and Okazaki fragment maturation pathway on fork stability upon stress. Specifically, it is demonstrated that the loss of PCNA ubiquitination leads to nascent DNA degradation by DNA2. The PCNA ubiquitination seems to be epistatic with ligation of Okazaki fragments (followed via LIG1) and PCNA unloading (followed via ATAD5). These results demonstrate that in the absence of proper completion of presumably lagging strand replication, upon stress, replication forks are targeted by DNA2.

Overall the experiments are well done. At points however, the conclusions seem to be somewhat correlative, so a revised manuscript should more rigorously support some of the claims. Also, the results are at times contradictory with previous literature (which in such a crowded field may not be too surprising), however, more effort should be done to clarify these differences.

Specific points:

1. The first confusing point is the discrepancy with the cited Vujanovic et al study. The Vujanovic paper shows that in wild types forks slow down and are reversed, which required PCNA ubiquitination and ZRANB3 translocase (Fig. 1B). In the current study, the authors show that in wild type cells forks are not affected, but are instead shortened in the PCNA KR cells (Fig. 2A). The figures appear to show exactly the opposite. The authors should clarify this difference.
2. The other important point is whether forks reverse. The authors use a correlative experiment: depletion of RAD51 restored nascent tract integrity in KR cells (Fig. 3A). Given the complications in fiber data interpretation, the authors should provide EM data (or collaborate with one of the labs where this is well established) to conclude that replication fork reversal occurs in KR cells. This is also important with regard to the Vujanovic story, where fork reversal was concluded to be impaired in KR cells.
3. Does UBC13 belong to the same pathway as another component of the PCNA-LIG1-ATAD5 axis?
4. There are rumors that the DNA2 inhibitor may not be entirely specific. The authors should include at least one control experiment with siDNA2.
5. The data in Fig. 4B seem again only correlative to indicate the Okazaki fragment ligation is compromised in KR cells. Can the authors test if the KR mutation is epistatic with siLIG1 in the comet assay that is more direct?

Minor point:

- please clarify or rephrase "okazaki fragment synthesis" in abstract

Reviewer #3 (Remarks to the Author):

In their manuscript, Thakar and co-workers describe how PCNA is involved in the protection of stalled replication forks from DNA2-mediated degradation.

Over the recent years, stalled replication forks have been shown to be protected from nucleolytic degradation. Fork protection has been linked to resistance to chemotherapeutic agents and involved multiple nucleases (including mre11, DNA2 and mus81) and a growing list of protective factors, including HR/Fanconi factors.

In this very well written manuscript, the role of PCNA ubiquitylation in fork protection is demonstrated. Moreover, the degradation was linked to defective gap-filling of the replication fork, and incomplete Okazaki fragment synthesis. These extensive results provide new insights into the protection of nascent tract DNA and the function of PCNA herein.

specific comments:

The authors switch back and forth between RPE1 and 293T (and occasionally U2OS) cells. Unclear what the rationale is to use which cell line.

The term: 'ubiquitination-deficient' in the abstract and for instance on page 6 suggests that this mutant cell line cannot perform ubiquitination at all. I would consistently use the name of the PCNA mutation.

Figure 1F: the amounts of foci in the example pictures does not do justice to the quantification, in which only a very minor difference is observed.

Figure 1: UV and or cisplatin sensitivity should also be tested for the RPE1 cell line.

Figure 2E: the effects of C5 are tested solely in the context of HU. Would be good to also test in absence of HU, to see if DNA2 is also involved in endogenous replication stress, like is also done in 2C, D.

Figure 4B: the experiment showing PAR chains is not explained to well in the results section. Also, the order of conditions make this experiment hard to interpret,

The results are coined as demonstrating a 'PCNA-LIG1-ATAD5-Caf1 pathway'. However, these genes are described only in a genetic way, rather than a biochemical pathway.

fork protection should be analysed upon Fen1 inactivation in wt/KR cells

The olaparib sensitivity assays are not very developed. Brca1 and Brca2 are typically essential in many (cancer) cell lines. These differences are not visible because the data is normalized. Would be good to show non-normalized data as well and include representative images of these clonogenic assays, in confirm these data using the other components of the pathway is identified. The olaparib sensitivity should be extended to cancer cells lines. Also, I am not sure what actually happens to these cells ? the authors should do a better job in characterising the lesions that accumulate in the combined BRCA/PCNA mutant setting.

I think the manuscript would be better if a model (simplified version of the model in the supplemental data) was included in the main figure.

Response to referees

We would like to thank the reviewers for their constructive comments. To address the reviewer's concerns, we are submitting a substantially revised manuscript with 25 new figure panels, as well as 4 revised figure panels and 4 new figures for reviewers. Below, please find our point-by-point reply to reviewers' comments (**our responses in red font**)

Reviewer #1

We are glad that the reviewer found that our findings “are very interesting and reveal a novel role for PCNA in maintenance of genomic integrity”, and that we “convincingly show the impact of PCNA mono-ubiquitination on maturation of Okazaki fragments and chromatin assembly”. We thank the reviewer for their helpful comments. We have addressed these comments in the revised manuscript as indicated below:

In spite of these strengths of the study, there are inconsistencies in the way the experiments are performed to examine the impact on fork degradation. The authors have switched between using fiber track length and ratio of Idu and CldU to measure fork degradation (For example, in Figure 3A and 3B CldU/IdU length is measured but in figure 3C IdU track length is measured). While both methods are correct, the authors should have used the same method in both cell lines throughout the study for consistency. Another concern is that the effects on fork protection are very subtle. Given the subtle effects of KR mutation on the ratio of DNA fiber track lengths, it is even more important to perform all studies in a similar way and under identical conditions.

We thank the reviewer for this comment. As the reviewer pointed out, both methods for quantifying HU-induced nascent tract degradation are correct. To strengthen the rigor of the study, we think it is important that, for the K164R mutant cell lines, we show nascent tract degradation using both methods, in both cell lines (Fig. 2a-d). However, we do agree with the reviewer that the fork degradation studies with gene knockdown were not performed consistently in the previous version of the manuscript. Thus, we repeated all these experiments using the CldU/IdU ratio setup, and included them in the revised manuscript (**Fig. 3c, Supplementary Fig. 2f, 2i, 2k, 3e, and 4i**) to replace the experiments performed under different conditions.

Some of the minor concerns are:

1. Figure 2C, D: C5 and Mirin treatment in WT cells is missing

We did not include the C5 and mirin conditions in these figures, as HU treatment does not induce nascent strand degradation in WT cells, and thus treatment with nuclease inhibitors is not expected to show any impact. We are including here a figure for the reviewer confirming that indeed, C5 and mirin do not impact fork stability upon HU treatment in wildtype cells.

2. *Figure S2F: Why is the CldU track length significantly reduced in HU treated 293T-WT cells compared to untreated cells? There should be no significant change (as shown in Fig S2D in RPE cells).*

The experimental setup used in the original Figure S2F was CldU-HU(4mM for 5h)-IdU. In contrast, in the original Figure S2D (renumbered as Supplementary Fig. 2d in the revised manuscript), as well as in Fig. 2a and 2b, the experimental setup used was CldU-HU(4mM for 3h)-IdU. The increased HU treatment time (5h vs 3h) explains why HU-induced degradation was observed in wildtype cells in the original Figure S2F but not in Fig. 2a, 2b, and Supplementary Fig. 2d. This is in line with a previous publication by the Vindigni group (PMID: 25733713) showing that prolonged incubation (6h and 8h) with 4mM HU results in nascent strand degradation in wildtype cells. In any case, in the revised manuscript, we replaced the original Figure S2F with the new **Supplementary Fig. 2f** in which nascent tract degradation was measured using the CldU/IdU ratio setup, in order to show all fork degradation data consistently under identical conditions, as requested by the reviewer in their main comment (see above).

3. *Figure S2Q and R and Figure 4D: What is level of Ub-PCNA in RAD18 KD and KO cells?*

The levels of PCNA ubiquitination in RAD18-deficient cells is presented in the new **Supplementary Fig. 4e and 4f**.

4. *Figure S2S: P values are switched.*

We apologize for this error, and thanks the reviewer for pointing it out. We corrected the error in the revised manuscript, in which the figure is renumbered as Supplementary Fig. 2o.

5. *Figure S2T, RPA foci are not visible. Signal is very diffused.*

We now provide a revised figure (renumbered as Supplementary Fig. 2s in the revised manuscript) in which we enhanced the contrast to increase the visibility of the foci in the representative micrographs shown. As described in the Methods section, foci quantification for all immunofluorescence experiments was performed automatically using ImageJ software.

6. *Figure S4G: No effect of REV1 KD in WT RPE cells is surprising. Since REV1 is known to promote PCNA mono-ubiquitination, which is dependent on RAD18 (Wang et al, 2016, J Cell Sc.), why is the IdU track length reduced in the absence of HU. Based on the impact of K164R mutation on track length in normal cells (Fig 2B), the IdU track length should have increased in REV1 KD cells. What is the level of mUb-PCNA in the REV1KD cells.*

We thank the reviewer for pointing out this issue, which we investigated as requested. We now show in the new **Supplementary Fig. 4j** that in our hands, knockdown of REV1 does not reduce PCNA ubiquitination (if anything, it is increased). While we do not know the cause of the discrepancy with the Wang et al report, this result explains why we cannot detect nascent tract degradation in REV1-depleted cells. In the revised manuscript, we further confirmed the absence of HU-induced nascent tract degradation in REV1-depleted cells by employing the

alternative fiber setup, using CldU/IdU ratio measurements. We replaced the original Figure S4G with the new **Supplementary Fig. 4i** using this setup, in order to show all fork degradation data consistently under identical conditions, as requested by the reviewer in their main comment (see above).

7. *Figure 4C: Average IdU track lengths are missing in this panel. The impact of KR mutation on IdU track length (in siControl, no HU) here is very modest compared to the effects seen in Figure 1A.*

We added the IdU tract lengths averages to this figure (renumbered as Fig. 4d in the revised manuscript). While the tract length increase in K164R cells may be more modest in this particular experiment compared to that shown in Fig. 2a, this simply reflects experimental variation. We have consistently observed a ~30% increase in K164R cells in many independent experiments, in both cell lines.

8. *Figure 6C: no error bar for WT siBRCA1 at 1 μ M olaparib. Why do 293T-KR cells exhibit no sensitivity to olaparib. Yet, when combined with BRCA1/2 loss, KR cells exhibit a significant increase in sensitivity compared to BRCA1/2 loss alone.*

The error bars are too small to be observable for that particular time point. The individual measurements are presented in the Source Data file.

In light of the role of PCNA ubiquitination in regulating Okazaki fragment maturation we describe in our manuscript, it is indeed intriguing that PCNA-K164R cells are not sensitive to olaparib, as recent work by the Caldecott lab (Hanzlikova et al, PMID: 29983321- ref49 in our manuscript) described a role for PARP1 in the detection and resolution of unligated Okazaki fragments. As the ssDNA gaps accumulating in PCNA-K164R during lagging strand synthesis interfere with Okazaki fragment maturation, these cells should be sensitive to PARP1 inhibition. To address this conundrum, we investigated the resolution of ssDNA gaps at later time points. We now show in the new **Fig. 6c** that, even in PCNA-K164R cells, these gaps are relatively short-lived as they can be efficiently repaired by the BRCA pathway, explaining why PCNA-K164R cells are not sensitive to olaparib. In contrast, in BRCA-deficient cells, the ssDNA gaps accumulating in the absence of PCNA ubiquitination are not properly resolved, accounting for the increased olaparib sensitivity upon concomitant inactivation of PCNA ubiquitination and the BRCA pathway. This model is carefully described in the last paragraph of the Discussion section of the revised manuscript.

Reviewer #2

We are glad that the reviewer found that the experiments shown in our manuscript “are well done” and “demonstrate that in the absence of proper completion of presumably lagging strand replication, upon stress, replication forks are targeted by DNA2”. We thank the reviewer for their helpful comments. We have addressed these comments in the revised manuscript as indicated below:

Specific points:

1. The first confusing point is the discrepancy with the cited Vujanovic et al study. The Vujanovic paper shows that in wild types forks slow down and are reversed, which required PCNA ubiquitination and ZRANB3 translocase (Fig. 1B). In the current study, the authors show that in wild type cells forks are not affected, but are instead shortened in the PCNA KR cells (Fig. 2A). The figures appear to show exactly the opposite. The authors should clarify this difference.

We apologize for this confusion, but the experiments pointed out by the reviewer are performed under different conditions, thereby explaining the different results: Figure 1B in the Vujanovic paper (PMID: 28886337) investigates fork slowing in response to DNA damaging agents, which are added at the same time with the second label (IdU) -see schematic in Vujanovic et al, Figure 1A. In contrast, the experiment we present in our manuscript in Fig. 2a is designed to measure HU-induced nascent tract degradation, and thus an acute concentration of the drug which completely abolishes fork progression is added in between the labels IdU and CldU. In fact, in our manuscript we acknowledge the Vujanovic results, and we present in the **Supplementary Fig. 2c** an experiment performed using a similar labeling scheme as in the Vujanovic paper (using a drug dose that impairs but does not arrest fork progression). This experiment shows the same result as in the Vujanovic paper, namely that fork slowing is reduced in K164R cells. Thus, there is no discrepancy between our findings and those in the Vujanovic paper. In the revised manuscript, we carefully explained this.

2. The other important point is whether forks reverse. The authors use a correlative experiment: depletion of RAD51 restored nascent tract integrity in KR cells (Fig. 3A). Given the complications in fiber data interpretation, the authors should provide EM data (or collaborate with one of the labs where this is well established) to conclude that replication fork reversal occurs in KR cells. This is also important with regard to the Vujanovic story, where fork reversal was concluded to be impaired in KR cells.

In our DNA fiber combing experiments, depletion of ZRANB3 only partly rescues nascent tract degradation in K164R cells (Fig. 3b). On one hand, this is in line with the Vujanovic paper, showing that ZRANB3 is important for fork reversal. On the other hand, it is also consistent with previous literature (Ciccia et al, PMID: 22704558) showing that ZRANB3 can interact directly with unmodified PCNA, through its PIP and APIM motifs. Thus, in PCNA-K164R cells, an interaction with PCNA, and a reduced but not absent fork reversal activity by ZRANB3, is not unexpected. We agree with the reviewer that electron microscopy visualization of reversed fork structures is a powerful technique which can uncover important mechanistic insights. However, this assay is, to our knowledge, currently established in only 2 or 3 laboratories in the world. We have contacted Dr. Alessandro Vindigni, one of these experts, and he indicated that these experiments are difficult, time consuming, and labor intensive, and are not feasible within the time frame of this revision. Thus, while we understand the importance of investigating fork reversal by EM as mentioned by the reviewer, this is outside the scope of the current manuscript.

To address the reviewer's comment, in the revised manuscript, we provide additional evidence that ZRANB3 fork reversal activity is still present in PCNA-K164R cells. Upon performing the UBC13 experiments requested by the reviewer in their comment #3, we found that UBC13 depletion also results in nascent tract degradation (see our answer to reviewer's comment #3 below). This finding provided us with an additional tool to investigate the impact of PCNA ubiquitination on ZRANB3 activity, as UBC13 is essential for PCNA K63-linked poly-ubiquitination. We found that ZRANB3 knockdown partially suppresses nascent tract

degradation in UBC13-depleted cells, similar to its impact in PCNA-K164R cells (new **Supplementary Fig. 3c**). This finding further supports our model that ZRANB3-mediated fork reversal is not abolished in the absence of PCNA ubiquitination.

In addition, we show that depletion of SMARCAL1 completely rescues nascent strand degradation in K164R cells (Fig. 3b). As SMARCAL1 is a DNA translocase with demonstrated activity in fork reversal, this result is a very strong indication that fork reversal is indeed required for nascent tract degradation in these cells. We would like to also point out that even in the Vujanovic paper, fork reversal was not abolished in K164R cells (in fact, under no drug treatment conditions, fork reversal was rather increased in K164R cells compared to control cells -Fig 1D). It is possible that this residual fork reversal is catalyzed by SMARCAL1.

Thus, overall our results suggest the model that in K164R cells, fork reversal is impaired (because of reduced ZRANB3 function) but not abolished (as SMARCAL1 is still active). In our revised manuscript, we are carefully explaining this model. However, we also discuss the possibility that other DNA structures formed by SMARCAL1 at stalled replication forks represent the relevant substrate of DNA2 in K164R cells.

3. Does UBC13 belong to the same pathway as another component of the PCNA-LIG1-ATAD5 axis?

We thank the reviewer for this suggestion. We now show in the new **Supplementary Fig. 2p-r** that, relatively unexpectedly, loss of UBC13 also results in DNA2-mediated nascent tract degradation. We speculate in the Discussion section that this finding may reflect a previously-proposed role for UBC13 in TLS (PMID: 20453833), or alternatively an involvement of template switching mediated by PCNA poly-ubiquitination in gap filling, or of other UBC13 substrates.

4. There are rumors that the DNA2 inhibitor may not be entirely specific. The authors should include at least one control experiment with siDNA2.

In the revised manuscript, we now show in the new **Supplementary Fig. 2k** that DNA2 knockdown by siRNA also suppresses HU-induced nascent strand degradation in K164R cells, similarly to treatment with the DNA inhibitor C5.

5. The data in Fig. 4B seem again only correlative to indicate the Okazaki fragment ligation is compromised in KR cells. Can the authors test if the KR mutation is epistatic with siLIG1 in the comet assay that is more direct?

We attempted to perform the experiment indicated by the reviewer. However, in our hands, LIG1 depletion in wildtype cells did not increase the percent tail DNA in the BrdU alkaline comet assay. We present this result here as figure for the reviewer. It is possible that the BrdU alkaline comet assay can only detect ssDNA gaps such as those present in K164R cells, but not ssDNA nicks which accumulate in the absence of Okazaki fragment ligation upon LIG1 depletion. This would explain why LIG1 depletion does not increase the percent tail DNA in this assay.

In any case, we would like to point out that the findings presented in Supplementary Fig. 4b using emetine, an inhibitor of lagging strand synthesis, indicate that the PAR chains detected in K164R cells are generated during lagging strand synthesis, therefore providing further evidence that Okazaki fragment maturation is compromised in K164R cells.

Minor point:

- please clarify or rephrase "okazaki fragment synthesis" in abstract

In the revised manuscript, we replaced this with "Okazaki fragment maturation".

Reviewer #3

We are glad that the reviewer found that our manuscript is "very well written" and our results "provide new insights into the protection of nascent tract DNA and the function of PCNA herein." We thank the reviewer for their helpful comments. We have addressed these comments in the revised manuscript as indicated below:

specific comments:

The authors switch back and forth between RPE1 and 293T (and occasionally U2OS) cells. Unclear what the rationale is to use which cell line.

In order to strengthen the rigor of the study, we think it is important that that we characterize the K164R mutation in both 293T and RPE1 cells. This is why, we show all important phenotypes in both cell types (eg. DNA damage sensitivity in Figure 1, fork protection defect in Figure 2 etc). For functional studies with gene knockdown, we generally present experiments performed in 293T cells, as we first created the K164R mutation in this background and thus we had it available for experiments for a longer time. However, the important findings were independently replicated in RPE1 cells. As the RPE1 cells are more amenable to siRNA-mediated gene knockdown, we employed these cells for some experiments where the level of gene knockdown was more critical (eg FEN1 depletion). We used the U2OS cells for a single experiment in Fig. 4e, simply because we already had the RAD18-knockout in this background available in the lab, and we wanted to increase the rigor of the study by showing the same phenotype in multiple cell line backgrounds.

The term: 'ubiquitination-deficient' in the abstract and for instance on page 6 suggests that this mutant cell line cannot perform ubiquitination at all. I would consistently use the name of the PCNA mutation.

We thank the reviewer for this comment, and made the indicated changes in the revised manuscript.

Figure 1F: the amounts of foci in the example pictures does not do justice to the quantification,

in which only a very minor difference is observed.

We replaced the immunofluorescence pictures in this figure (which was renumbered as Fig. 1h in the revised manuscript) to more accurately reflect the quantification presented in the graph.

Figure 1: UV and or cisplatin sensitivity should also be tested for the RPE1 cell line.

We now show in the new **Fig. 1c and 1e** that RPE1-K164R cells are sensitive to UV and cisplatin, similar to 293T-K64R cells.

Figure 2E: the effects of C5 are tested solely in the context of HU. Would be good to also test in absence of Hu, to see if DNA2 is also involved in endogenous replication stress, like is also done in 2C, D.

Treatment of wildtype cells with C5 results in increased 53BP1 foci formation (see figure for the reviewer). We have not included these results in the text, as they are in line with previously reported findings by the Stewart lab (PMID: 22570476) that DNA2 depletion in wildtype cells results in spontaneous DNA damage as measured by γ H2AX formation. In contrast, we show in our manuscript that the HU-induced DNA damage is suppressed by DNA2 inhibition in K164R cells (Fig. 2e and 2f).

Figure 4B: the experiment showing PAR chains is not explained to well in the results section. Also, the order of conditions make this experiment hard to interpret,

We apologize for this. We now include a more detailed description of this figure (which was renumbered as Fig. 4c in the revised manuscript).

The results are coined as demonstrating a 'PCNA-LIG1-ATAD5-Caf1 pathway'. However, these genes are described only in a genetic way, rather than a biochemical pathway.

We agree with reviewer, and in the revised manuscript took care to describe this as a genetic pathway, throughout the text.

Fork protection should be analysed upon Fen1 inactivation in wt/KR cells

We thank the reviewer for this suggestion. We now show in the new **Fig. 4b and Supplementary Fig. 4d** that, similarly to LIG1 knockdown, depletion of FEN1 also results in increased replication tract length under normal conditions, and HU-induced nascent tract degradation, which are epistatic to PCNA-K164R. These findings provide further evidence that Okazaki fragment maturation and PCNA ubiquitination operate in the same fork protection pathway.

The olaparib sensitivity assays are not very developed. *Brca1* and *Brca2* are typically essential in many (cancer) cell lines. These differences are not visible because the data is normalized. Would be good to show non-normalized data as well and include representative images of these clonogenic assays, in confirm these data using the other components of the pathway is identified. The olaparib sensitivity should be extended to cancer cells lines. Also, I am not sure what actually happens to these cells? the authors should do a better job in characterising the lesions that accumulate in the combined BRCA/PCNA mutant setting.

Under the assay conditions employed (transient siRNA transfection), we do observe a reduction in plating efficiency upon BRCA2-depletion (but not upon BRCA1-depletion). We present here a figure for the reviewer illustrating this genetic interaction. To circumvent this, in our clonogenic assays we adjusted the number of cells plated for the conditions with lower plating efficiency. In the revised manuscript, we are presenting representative images of the clonogenic assays as the new **Supplementary Fig. 6c and 6d**, and the non-normalized data for Fig. 6d and 6e in the Source Data file. As mentioned above, since the number of cells plated was adjusted for plating efficiency, these figures do not reflect the reduced viability of BRCA2-depleted cells.

As requested by the reviewer, in the revised manuscript we extended these analyses to HeLa cancer cell lines, and to other members of the pathway, including RAD18 and ATAD5 (new **Fig. 6f and 6g**). In all cases, concomitant inactivation of the PCNA-ubiquitination and the BRCA pathways enhances olaparib sensitivity.

Finally, as requested by the reviewer, we characterized the lesions that accumulate in the double mutant. Using the BrdU alkaline comet and the neutral comet assays, we found that replication-dependent ssDNA gaps accumulate equally in BRCA2-proficient and deficient KR cells during DNA synthesis; however, 5h after DNA synthesis, these gaps are repaired in BRCA2-proficient cells, but are still present in the BRCA2-depleted cells (new **Fig. 6c, Supplementary Fig. 6a**). These findings indicate that loss of BRCA2 impairs the repair of ssDNA gaps accumulating in KR cells under normal DNA replication conditions.

I think the manuscript would be better if a model (simplified version of the model in the supplemental data) was included in the main figure.

We thank the reviewer for this suggestion. We now included a simplified version of the model as the new **Fig. 5e** in the revised manuscript.

REVIEWERS' COMMENTS:

Reviewer #1 (Remarks to the Author):

The authors have very thoroughly and carefully addressed all my concerns. I am satisfied with the changes.

Reviewer #2 (Remarks to the Author):

The authors clarified a number of points, which improved the manuscript. I agree that the EM analysis would be above the scope, and the authors' answers and additional data clarify the point.

Reviewer #3 (Remarks to the Author):

The authors have done a good job in responding to my comments.

one minor comment that I have is about line 427: '...contributes to PARPi resistance in these cells.'

I would change this in '...determines PARPi sensitivity.'

PCNA ubiquitination has an effect on toxicity of PARP inhibitors in BRCA-deficient cells, but I would not use the phrase 'PARPi resistance', as this term is used for mechanisms that reverse observed PARP inhibitor sensitivity in BRCA mutant settings.

Response to referees

We are glad the reviewers were satisfied with our manuscript revision.

Reviewer #1

The authors have very thoroughly and carefully addressed all my concerns. I am satisfied with the changes.

Reviewer #2

The authors clarified a number of points, which improved the manuscript. I agree that the EM analysis would be above the scope, and the authors' answers and additional data clarify the point.

Reviewer #3

The authors have done a good job in responding to my comments. one minor comment that I have is about line 427: '...contributes to PARPi resistance in these cells.' I would change this in '...determines PARPi sensitivity.' PCNA ubiquitination has an effect on toxicity of PARP inhibitors in BRCA-deficient cells, but I would not use the phrase 'PARPi resistance', as this term is used for mechanisms that reverse observed PARP inhibitor sensitivity in BRCA mutant settings.□

We agree with the reviewer and have modified this sentence as the reviewer indicated.